# Enforcing Idempotency in Neural Networks

**Nikolaj Banke Jensen** [1]  **Jamie Vicary** [2]

## Abstract

In this work, we propose a new architecture-agnostic method for training idempotent neural networks. An idempotent operator satisfies $f(\mathbf{x}) = f(f(\mathbf{x}))$, meaning it can be applied iteratively with no effect beyond the first application. Some neural networks used in data transformation tasks, such as image generation and augmentation, can represent non-linear idempotent projections. Using methods from perturbation theory we derive the recurrence relation $\mathbf{K}' \leftarrow 3\mathbf{K}^2 - 2\mathbf{K}^3$ for iteratively projecting a real-valued matrix $\mathbf{K}$ onto the manifold of idempotent matrices. Our analysis shows that for linear, single-layer MLP networks this projection 1) has idempotent fixed points, and 2) is attracting only around idempotent points. We give an extension to non-linear networks by considering our approach as a substitution of the gradient for the canonical loss function, achieving an architecture-agnostic training scheme. We provide experimental results for MLP- and CNN-based architectures with significant improvement in idempotent error over the canonical gradient-based approach. Finally, we demonstrate practical applications of the method as we train generative networks on MNIST and CelebA successfully using only a simple reconstruction loss paired with our method.

## 1. Introduction

Using neural networks as data augmentation tools is becoming more widespread in areas such as signal processing and generative artificial intelligence. In particular, networks of the form $f : X \to X$, mapping data within the same space $X$, are frequently used in image augmentation (Lu et al.,

[1]Department of Computer Science, University of Oxford, Oxford, UK. [2]Department of Computer Science and Technology, University of Cambridge, Cambridge, UK. Correspondence to: Nikolaj Banke Jensen <nikolaj.jensen@cs.ox.ac.uk>, Jamie Vicary <jamie.vicary@cl.cam.ac.uk>.

*Proceedings of the 42nd International Conference on Machine Learning*, Vancouver, Canada. PMLR 267, 2025. Copyright 2025 by the author(s).

2022), video generation (Ma et al., 2020; Liu et al., 2021), sorting algorithms (Tambouratzis, 1999), compression algorithms (Namphol et al., 1996; Liu et al., 2021), image denoising (Ilesanmi & Ilesanmi, 2021; Liu et al., 2021; Mao et al., 2023), and image generation (Liu et al., 2021), among others.

Some data transformation tasks admit *only* idempotent solutions (*e.g.*, sorting), whilst other tasks admit *no* idempotent solutions (*e.g.*, rotating an image by $90°$). This work is concerned with a class of data transformation tasks which has both idempotent and non-idempotent solutions and where idempotency might be a desirable property. For example, in Section 3 we study idempotency in generative networks where it is the formal requirement of one-step inference, but also denoising and image augmentation (*e.g.*, application of effect-filters) are examples of tasks where idempotent solutions may be desirable (Liu et al., 2021; Mao et al., 2023). Since solutions are not inherently idempotent in this class, we explore actively enforcing idempotency as a component of the loss function used in training.

In this paper we are primarily concerned with networks $f_{\boldsymbol{\theta}} : \mathbb{R}^n \to \mathbb{R}^n$, where $\boldsymbol{\theta}$ is a collection of weight parameters. The condition that $f_{\boldsymbol{\theta}}$ is idempotent is the following, for all $\mathbf{x} \in \mathbb{R}^n$:

$$f_{\boldsymbol{\theta}}(\mathbf{x}) = f_{\boldsymbol{\theta}}(f_{\boldsymbol{\theta}}(\mathbf{x})). \tag{1}$$

If $f_{\boldsymbol{\theta}}(\mathbf{x}) = \mathbf{W}\mathbf{x}$ (a single-layer, fully-connected network with no bias and the identity activation function) where $\mathbf{W} \in \mathbb{R}^{n \times n}$ is the weight matrix, then condition (1) reduces to the familiar notion from linear algebra where $\mathbf{W} = \mathbf{W}^2$ and eigenvalues of $\mathbf{W}$ are either 0 or 1. Condition (1) also gives the correct notion for non-linear networks acting as idempotent projections, and can be optimized using a simple mean-squared error loss, where $\mathbf{x} \in \mathbb{R}^n$:

$$\mathcal{L}_{\text{idem}}(\mathbf{x}) = \frac{1}{m} \sum_{i=1}^{m} \left( f_{\boldsymbol{\theta}}(f_{\boldsymbol{\theta}}(\mathbf{x})) - f_{\boldsymbol{\theta}}(\mathbf{x}) \right)^2. \tag{2}$$

As we show in Section 3, minimizing this loss using canonical gradient descent can yield relatively poor improvement in the idempotent loss. Additionally, due to the secondary application of $f_{\boldsymbol{\theta}}$ the number of terms in the gradient $\nabla_{\boldsymbol{\theta}} \mathcal{L}_{\text{idem}}$ grows exponentially in the number of layers if memoization is not used, making the approach computationally expensive for certain architectures. If memoization is

used, then this can be reduced to linear growth, as discussed in Section 3.3.

In this work, we propose an alternative method for training neural networks to satisfy condition (1). Using ideas from Perturbation Theory (Kato, 1995) we derive a function $g$ which solves $\mathbf{K}' = g(\mathbf{K})$ such that if $\mathbf{K} \in \mathbb{R}^{n \times n}$ is an "almost" idempotent matrix, then $\mathbf{K}' \in \mathbb{R}^{n \times n}$ is perfectly idempotent (*i.e.*, $(\mathbf{K}')^2 = \mathbf{K}'$). In this work, we focus on one such function:

$$g(\mathbf{K}) = 3\mathbf{K}^2 - 2\mathbf{K}^3. \tag{3}$$

Although we assume $\mathbf{K}$ is close to idempotent, we show that in practice $g$ can be used to derive matrices which are within machine precision of perfect idempotency even when the input matrix $\mathbf{K}$ is relatively far from idempotent. At a high level, this process is based on a recurrence relation $\mathbf{K}' = \mathbf{K} + \gamma(g(\mathbf{K}) - \mathbf{K})$, taking small $\gamma$-sized steps in the direction of $g(\mathbf{K})$. While this recurrence relation derives idempotent matrices—and can therefore be used to train single-layer networks with identity activations to be idempotent—we also give a more general application of Eq. (3) as a modification of the backpropagation algorithm, yielding an architecture agnostic and efficient algorithm for finding idempotent networks. As we will see, this modification in general not only leads to significantly improved idempotent error reduction but also explores the loss landscape differently from the canonical approach.

In Section 2.1 we give a detailed description of the method used to derive Eq. (3) and alternative solutions. We also show that while there exists non-idempotent fixed points to Eq. (3), these points are repelling under the recurrence relation $\mathbf{K}' = \mathbf{K} + \gamma(g(\mathbf{K}) - \mathbf{K})$ for $0 \leq \gamma \leq 1$, giving credence to the use of such a recurrence relation in practice. Finally, in Section 2.3 we derive a full training scheme for training arbitrary neural network architectures of the form $f_{\boldsymbol{\theta}} : \mathbb{R}^n \to \mathbb{R}^n$. In Section 3, we present experimental data for a variety of fully-connected network architectures, showing that our method outperforms ordinary backpropagation under varied conditions. We also replicate the results of Shocher et al. 2023 by applying our method on a U-net style DCGAN model to successfully create generative networks for the MNIST and CelebA datasets. Lastly, Sections 4 and 5 discuss how our method distinguishes itself from related approaches as well as future work.

## 2. Method

### 2.1. An idea from Perturbation Theory

Perturbation Theory comprises methods for finding an approximate solution to a problem by starting from the exact solution of a related, simpler problem and adding successive "perturbations" to the system. It is a diverse set of tools used

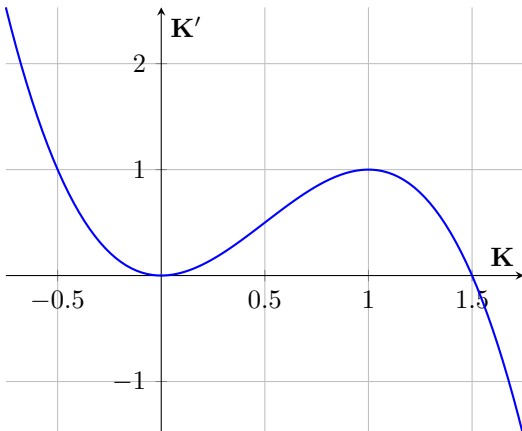

*Figure 1.* Plot of $\mathbf{K}' = 3\mathbf{K}^2 - 2\mathbf{K}^3$ in the case $\mathbf{K}$ is scalar.

to reason about complex dynamical systems often used in physics and quantum chemistry (Hirschfelder et al., 1964). We refer the reader to Kato 1995 for a detailed treatment of the topic.

We first define the term *near-idempotent* used throughout:

**Definition 2.1** (**Near-idempotent to order** $n$). Let the matrix $\mathbf{P} \in \mathbb{R}^{m \times m}$ satisfy $\mathbf{P} = \mathbf{P}^2$. Let $\mathbf{D} \in \mathbb{R}^{m \times m}$ be arbitrary (*e.g.*, noise) where there exists some $n \in \mathbb{N}$ such that $\mathbf{D}^{n+1}$ has coefficients with absolute value below $\epsilon \ll 1$. We say that $\mathbf{K} = \mathbf{P} + \mathbf{D}$ is **near-idempotent to order** $n$.

Using Definition 2.1 we may define the following ansatz in terms of a near-idempotent $\mathbf{K}$:

$$\mathbf{K}' = \alpha_1 \mathbf{K} + \alpha_2 \mathbf{K}^2 + \cdots + \alpha_j \mathbf{K}^j. \tag{4}$$

This poses $\mathbf{K}'$ as the linear combination of higher orders of near-idempotent matrices. If we further constrain $(\mathbf{K}')^2 - \mathbf{K}' = \mathbf{0}$, the result is a system of polynomial equations in variables $\alpha_i$. Importantly, for all equations of the system, any term in which $\mathbf{D}$ appears at least $n + 1$ times can be considered "negligible" and ignored. This simplification vastly reduces the problem and allows approximate solutions. The coefficients $\alpha_i$ can be thought of as parameterizing a projection $g$ such that $\mathbf{K}' = g(\mathbf{K})$ for an arbitrary near-idempotent $\mathbf{K}$. The requirement that $\mathbf{K}'$ be idempotent and that $\mathbf{K}$ is only near-idempotent implies that a solution $g$ is a projection onto the manifold of idempotent matrices; we call $g$ an **idempotent corrector** as it must "make $\mathbf{K}$ idempotent".

Note that Definition 2.1 places no restrictions on the distribution from which $\mathbf{D}$ is drawn, hence $\mathbf{K}$ and the underlying $\mathbf{P}$ have no presumed relation. Additionally, the equation 4 above also places no assumptions on the relationship between $\mathbf{P}$ and $\mathbf{K}'$.

In the case when $n = 1$ we consider $\mathbf{D}^2 \approx \mathbf{0}$ and the expression $(\mathbf{K}')^2 - \mathbf{K}' = \mathbf{0}$ can be expanded and reduced

by recursively applying the following assumptions, for all $\mathbf{X}, \mathbf{Y}, \mathbf{Z}$ matrices:

$$\mathbf{D}^2 \approx \mathbf{0}, \quad \mathbf{P}^2 = \mathbf{P}, \quad \mathbf{XDYDZ} \approx \mathbf{0}. \tag{5}$$

When $j \leq 2$, there exists no solutions for $\alpha_i$. When $j = 3$ there is exactly one solution when $\alpha_1 = 0$, $\alpha_2 = 3$ and $\alpha_3 = -2$, which gives precisely $g$ as defined in Eq. (3). For $j > 3$ there exists families of solutions (see Appendix A), but we consider primarily the case when $j = 3$ as this requires fewer higher-order terms of $\mathbf{K}$ and is therefore generally less costly to evaluate for concrete values. Note also that in general, solving the above system of polynomial equations is NP-hard or worse, but this is not a concern for us in practice as the number of variables $j$ is low (so all constraints have low degree also).

## 2.2. Fixed Points and Stability Analysis

Undoubtedly, a required property of any idempotent corrector $g$ is that every idempotent matrix is a fixed point, but it may also be desirable to find if any non-idempotent matrices are fixed points. Concretely, we wish to characterize solutions to $\mathbf{K} = 3\mathbf{K}^2 - 2\mathbf{K}^3$.

In general, we place no restrictions on the matrix $\mathbf{K} \in \mathbb{R}^{m \times m}$. In particular, it might not be directly diagonalizable. It is well known, however, that for every square matrix $\mathbf{K}$ there exists an invertible matrix $\mathbf{P}$ and a Jordan normal form (H. Weintraub, 2009) $\mathbf{J} \in \mathbb{C}^{m \times m}$ of $\mathbf{K} \in \mathbb{R}^{m \times m}$ such that $\mathbf{K} = \mathbf{PJP}^{-1}$. From this the dual problem,

$$\mathbf{J} = 3\mathbf{J}^2 - 2\mathbf{J}^3, \tag{6}$$

can be constructed. The block-diagonal structure of $\mathbf{J}$ imposes up to four equations per block of size $(k \times k)$ (see Appendix B):

$$\lambda = 3\lambda^2 - 2\lambda^3 \tag{7}$$
$$1 = 6\lambda - 6\lambda^2 \qquad \text{Only when } k \geq 2. \tag{8}$$
$$0 = 3 - 6\lambda \qquad \text{Only when } k \geq 3. \tag{9}$$
$$0 = 0 - 2 \qquad \text{Only when } k \geq 4. \tag{10}$$

Clearly, this system of equations is inconsistent when $k \geq 2$, hence algebraic multiplicity and geometric multiplicity of each eigenvalue have to be equal. This implies that $\mathbf{J}$ is diagonalizable for any fixed point $\mathbf{K}$. Furthermore, the solutions which satisfy only Eq. (7) are:

$$\lambda \in \{0, 0.5, 1\}. \tag{11}$$

Therefore, any fixed point of $\mathbf{K} = 3\mathbf{K}^2 - 2\mathbf{K}^3$ must have eigenvalues in this set. Consequently, all idempotent matrices are fixed points, but there exists also non-idempotent fixed points.

Although the initial derivation of $g(\mathbf{K}) = 3\mathbf{K}^2 - 2\mathbf{K}^3$ relies on $\mathbf{K}$ being near-idempotent to the first order, we consider more generally the behaviour of $g$ around the fixed points when applied repeatedly as a recurrence relation. Let $h(\lambda) = 3\lambda^2 - 2\lambda^3$ and observe its derivative $h'(\lambda) = 6\lambda - 6\lambda^2$. Then, for each fixed point of $g$ we have

$$h'(0) = 0, \quad h'(0.5) = 1.5, \quad h'(1) = 0. \tag{12}$$

Since $|h'(\lambda)| < 1$ for $\lambda \in \{0, 1\}$ these points are attracting whilst $|h'(\lambda)| > 1$ for $\lambda = 0.5$, thus this point is repelling.

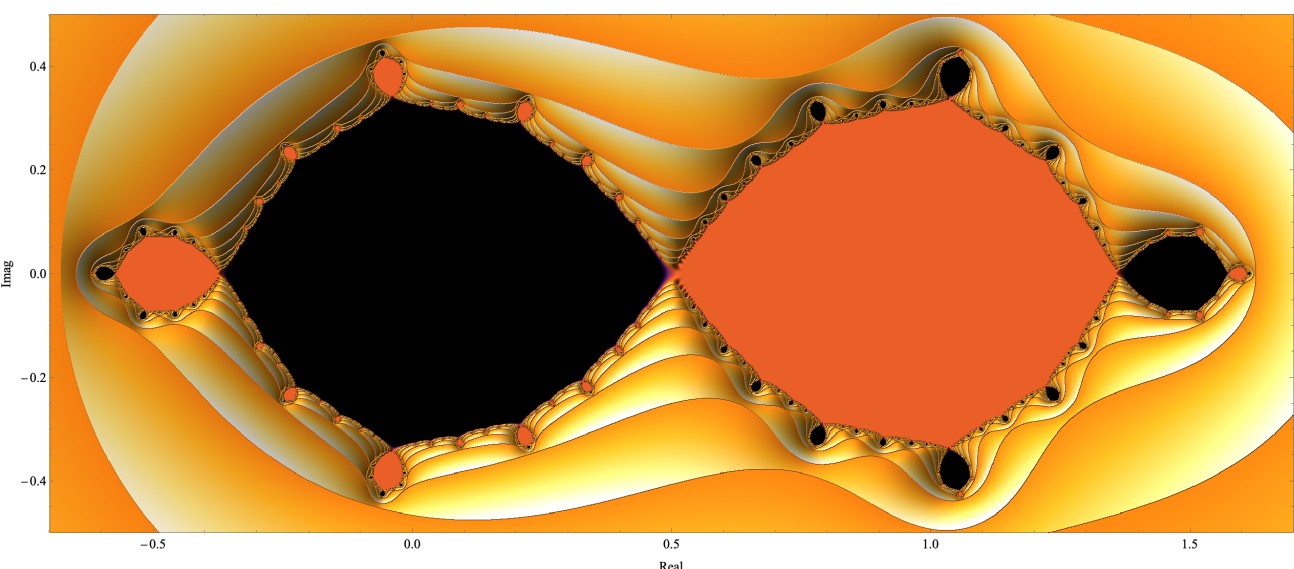

Figure 2. 10-time recursive application of $h(\lambda) = 3\lambda^2 - 2\lambda^3$ for each point on the complex plane. Black areas denote points converging onto 0, while orange areas denote points converging onto 1.

In other words, if the idempotent corrector $g$, applied as a recurrence relation on $\mathbf{K}$, converges at some point $\mathbf{K}'$, then $\mathbf{K}'$ will be approximately idempotent unless $\mathbf{K}$ has an eigenvalue of exactly $0.5$.

Furthermore, Figure 2 shows the result of applying the idempotent corrector recursively 10 times for each point on the complex plane. The attracting regions around $0$ and $1$ are large, hence any matrix that is "reasonably close" to idempotent will be projected onto a (within machine precision) idempotent matrix.

Whilst this analysis technically only applies in the linearized setting, we propose to also apply the method in non-linear settings using the following recurrence relation, for $0 \leq \gamma \leq 1$:

$$\mathbf{K}' = \mathbf{K} + \gamma(g(\mathbf{K}) - \mathbf{K}). \tag{13}$$

This has the effect of taking small $\gamma$-sized steps in the direction of $g$ at every time point.

### 2.3. Deriving a Training Scheme

Gradient-based optimization techniques use the gradient of an often non-convex loss function as the directional information used to update the hypothesis at each time step. This highlights a core difference between our approach and conventional gradient-based approaches, since the recurrence relation derived above (and shown in Figure 1) exactly describes the "direction" to move in to reduce idempotent error. Our method need only *evaluate* $g$ – finding its derivative is unimportant.

Consider a neural network $f_{\boldsymbol{\theta}} : \mathbb{R}^m \to \mathbb{R}^m$ together with its application to input $\mathbf{x} \in \mathbb{R}^m$, denoted $\mathbf{y} = f_{\boldsymbol{\theta}}(\mathbf{x})$. We might then consider the recurrence relation in Eq. (3) in the following form:

$$\mathbf{y}' = 3f_{\boldsymbol{\theta}}(\mathbf{y}) - 2f_{\boldsymbol{\theta}}(f_{\boldsymbol{\theta}}(\mathbf{y})) \tag{14}$$

This describes a desired change in the output of the network which we denote $\Delta f_{\boldsymbol{\theta}}(\mathbf{x}) = \mathbf{y}' - \mathbf{y}$. In other words, $\Delta f_{\boldsymbol{\theta}}(\mathbf{x})$ describes the change in $\mathbf{y}$ which moves $\mathbf{y}$ towards an idempotent projection much in the same way that the quantity $\frac{\partial(-\mathcal{L}_{\text{idem}}(\mathbf{y}))}{\partial \mathbf{y}}$ describes the direction which reduces the idempotent loss function in Eq. (2). A central idea presented in this work is therefore the definition

$$\frac{\partial(-\mathcal{L}_{\text{idem}}(\mathbf{y}))}{\partial \mathbf{y}} \equiv \Delta f_{\boldsymbol{\theta}}(\mathbf{x}) \tag{15}$$

as an alternative quantity to the traditional, analytical solution to $\frac{\partial(-\mathcal{L}_{\text{idem}}(\mathbf{y}))}{\partial \mathbf{y}}$. To complete the scheme, we consider how a change in the output $\mathbf{y}$ can be propagated to a change in the parameters $\boldsymbol{\theta}$ of $f_{\boldsymbol{\theta}}$. This, however, is a straightforward application of the chain rule as it is calculated conventionally in backpropagation. In this paper we use the term "**Modified Backpropagation**" to refer to the canonical backpropagation algorithm with the rule (15) applied appropriately when computing gradients.

One way to understand why this approach is sensible is to consider that in the linear case we obtain exactly the directional information $(3\mathbf{K}^2 - 2\mathbf{K}^3 - \mathbf{K})$ of Eq. 13 from the previous section. In the case when $f_{\boldsymbol{\theta}}$ is non-linear we wish for the network to act in an idempotent way around inputs taken from the training distribution with the expectation that enough such points yields idempotent behaviour for the rest of the distribution. We can approximately achieve this by enforcing the idempotency of the Jacobian $J_{\boldsymbol{\theta}}$ at $\mathbf{x}$. In our scheme this would give the objective

$$(3J_{\boldsymbol{\theta}}(\mathbf{x})^2 - 2J_{\boldsymbol{\theta}}(\mathbf{x})^3 - J_{\boldsymbol{\theta}}(\mathbf{x}))\mathbf{x} \tag{16}$$

which can be seen exactly as the linearized counterpart to $\Delta f_{\boldsymbol{\theta}}(\mathbf{x})$. Therefore, under the assumption that $f_{\boldsymbol{\theta}}$ behaves locally linearly we should expect the training scheme presented in this section to also optimize for idempotency in the non-linear setting at least around the training samples.

In practice, the definition (15) can be implemented in common machine learning frameworks, such as Jax and PyTorch as a user-defined automatic differentiation rule (see Appendix C).

## 3. Experimental Results

To evaluate the training scheme suggested in Section 2.3 we compare relative performance between the two methods: "Ordinary Backpropagation" with the quantity $\frac{\partial(-\mathcal{L}_{\text{idem}}(\mathbf{y}))}{\partial \mathbf{y}}$ resolved at runtime by automatic differentiation, and "Modified Backpropagation" with the modified backpropagation rule for $\frac{\partial(-\mathcal{L}_{\text{idem}}(\mathbf{y}))}{\partial \mathbf{y}}$. To demonstrate the flexibility of the approach, we report results for four diverse MLP-style networks, as described in Table 1.

The dataset used for training in this section is drawn from a normal distribution with mean $0$ and standard deviation $1$. To prevent concerns about overfitting, the distribution is sampled i.i.d. at each epoch during training. Furthermore, a batch size of $1000$ is used, although comparable results have been found for batch sizes between $32$ and $10\,000$. The optimizer used is SGD.

### 3.1. Qualitative Differences

In this section we present suggestive evidence that Modified Backpropagation searches the solution space differently from Ordinary Backpropagation.

For purposes of visualization, we employ the methods of Li et al. 2018 to compare the optimizer trajectories of Modified Backpropagation and Ordinary Backpropagation. Concretely, we train a copy of the same network with either

*Table 1.* Four neural networks for testing. Each "Linear($n$, $m$)" block is parameterized by its input dimension $n$ and its output dimension $m$, corresponding to the underlying $\mathbf{W} \in \mathbb{R}^{m \times n}$ weight matrix. Every block has an associated bias vector and LeakyReLU(0.2) activation function. B1 represents a trivial network, B2 represents a relatively deep network, B3 represents a relatively wide network, and B4 represents a more realistic network.

| Identifier | Architecture | No. Parameters |
|:----------:|:-------------|:--------------:|
| B1 | Linear(5, 5) | 30 |
| B2 | Linear(128, 256) | 263 296 |
|    | Linear(256, 256) | |
|    | Linear(256, 256) | |
|    | Linear(256, 256) | |
|    | Linear(256, 128) | |
| B3 | Linear(4096, 1024) | 8 393 728 |
|    | Linear(1024, 4096) | |
| B4 | Linear(784, 1024) | 4 509 456 |
|    | Linear(1024, 2048) | |
|    | Linear(2048, 784) | |

algorithm and record model parameters $\boldsymbol{\theta}_t^{\mathrm{MB}}$ and $\boldsymbol{\theta}_t^{\mathrm{OB}}$ at epoch $t$. A PCA analysis is then performed over the relative change in parameters from $\boldsymbol{\theta}_0^{\mathrm{MB}}$ and $\boldsymbol{\theta}_0^{\mathrm{OB}}$ (which are identical), from which we select the two most explanatory directions. Lastly, the loss landscape and trajectory paths

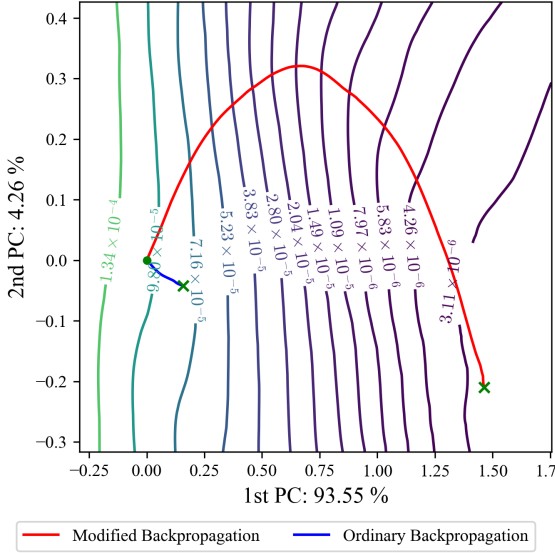

*Figure 3.* Representative projections of the optimizer trajectories over 2500 epochs of either algorithm on the B2 model at optimal learning rates (Figure 6). Total variance captured is $> 97.8\%$ with cosine similarity of PC1 and PC2 less than $1.0 \times 10^{-6}$. Optimizer trajectory of Modified Backpropagation deviates significantly from Ordinary Backpropagation.

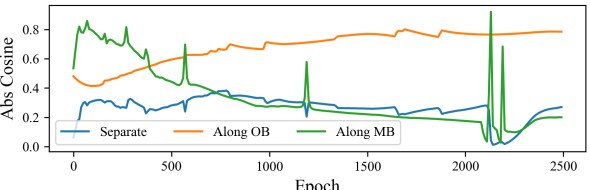

*Figure 4.* Absolute cosine similarity of gradients over time of a representative training run with model B2. "Along OB" optimizes the network with Ordinary Backpropagation and compares at each timepoint with suggested gradient from Modified Backpropagation. "Along MB" optimizes the network with Modified Backpropagation and compares with suggested gradient from Ordinary Backpropagation. "Separate" compares gradients of each optimizer as they independently optimize the network. Gradients suggested by Modified Backpropagation remains significantly different from those suggested by Ordinary Backpropagation.

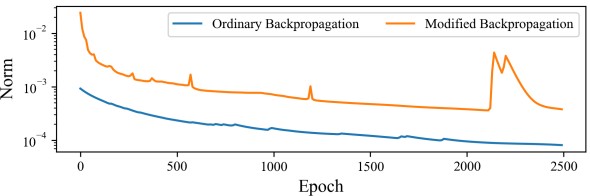

*Figure 5.* Norm of gradients over time of a representative training run with model B2. The network is optimized independently by either algorithm at optimal learning rates (Figure 6). Modified Backpropagation gives consistently stronger gradient signal than Ordinary Backpropagation.

$\boldsymbol{\theta}_t^{\mathrm{MB}}$ and $\boldsymbol{\theta}_t^{\mathrm{OB}}$ are projected onto the selected dimensions. An example is shown in Figure 3 (and Appendix H).

Qualitative evaluation show that Modified and Ordinary Backpropagation often differ significantly in projected trajectories across the two most explanatory directions, but this is not always the case (*e.g.*, B4 in Figure 16). Additionally, optimization trajectories for Modified Backpropagation can be explained by projection onto two direction with more than 90% variance explained, indicating that it exhibits the same behaviour as Ordinary Backpropagation which has previously been suggested to largely operate in low-dimensional subspaces (Li et al., 2018; Song et al., 2024). One should note, however, that the loss surface is here represented under a dramatic dimensionality reduction which limits further conclusions.

We now investigate how the gradients produced by Modified Backpropagation differ from those produced by Ordinary Backpropagation. We give here an analysis over a single training run on network B2, but similar results hold for all networks in Table 1 over repetitions of the experiment. As Figure 4 shows, gradients suggested by either algorithm remain relatively dissimilar throughout training, which further

indicates a difference in the expected optimization trajectory. Furthermore, as evidenced by Figures 3 and 5, Modified Backpropagation travels faster (*i.e.*, gives stronger gradient signal) than Ordinary Backpropagation, even when optimal learning rates are selected for both algorithms.

### 3.2. Quantitative Differences

We now give an evaluation of the relative efficacy of Modified Backpropagation to Ordinary Backpropagation. As shown in Figure 6, for networks B2-B4 Modified Backpropagation achieves significantly lower absolute idempotent error on average at lower learning rate. For network B3 the difference is more than one order of magnitude. As

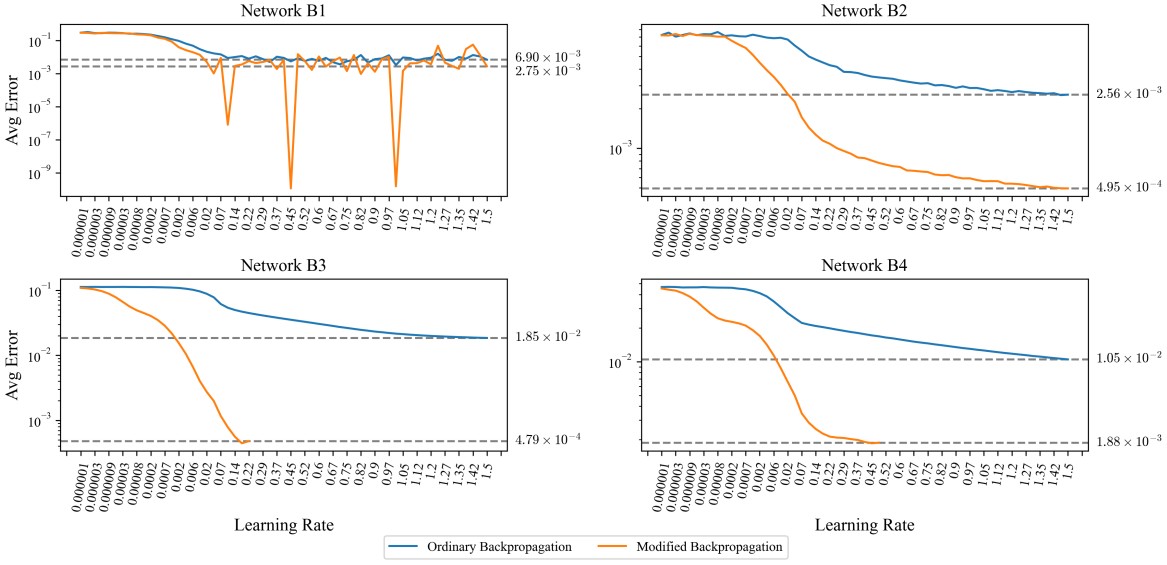

*Figure 6.* Average of 10 runs of each algorithm for a variety of learning rates. Networks are randomly initialized and trained for 2 500 epochs. Runs which did not return a network with lower idempotent error than the initial value are discarded, and the average is over remaining runs. For networks B3 and B4, learning rates $> 0.22$ and $> 0.52$ respectively had no runs with improvement in error. For Modified Backpropagation on B1, some runs resulted in approximately **0** which, due to floating-point imprecision, results in the error spikes.

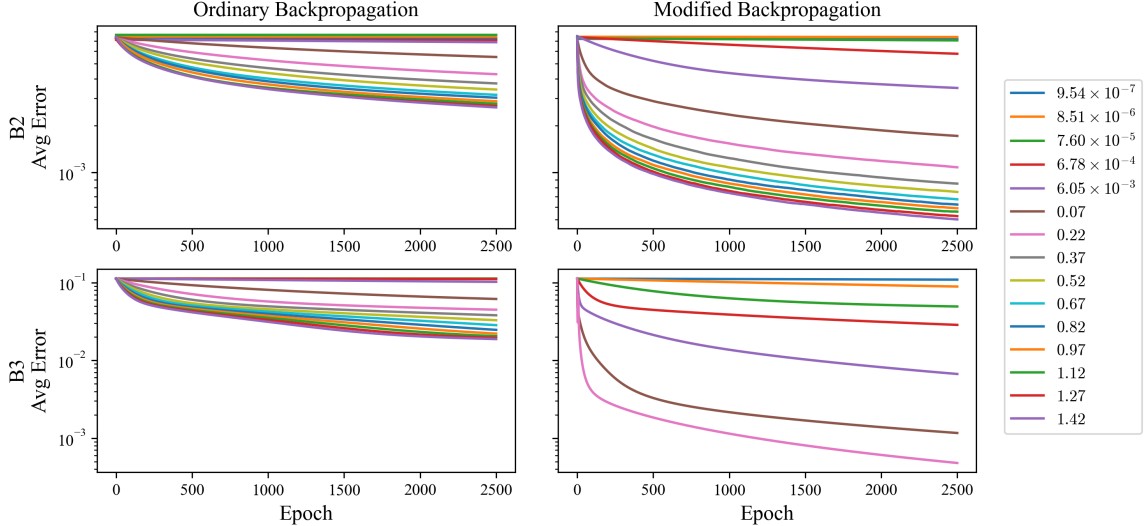

*Figure 7.* On networks B2 and B3, the average idempotent error across 10 runs for each learning rate is reported for each algorithm. Each column of graphs represents one algorithm. Modified Backpropagation achieves lower idempotent error at lower learning rates than Ordinary Backpropagation. The biggest relative improvement between algorithms occurs in the first $\sim 500$ epochs.

the tested networks represent varying architectures with a commonly used activation function, these results suggest that Modified Backpropagation fares well in a variety of training configurations.

Although the dataset used here is i.i.d. samples drawn from a Gaussian $\mathcal{N}(0,1)$, we observe similar results when data comes from other distributions, such as the uniform distribution $\mathcal{U}(-k,k)$ for $k \in \mathbb{N}$. Following Shocher et al. 2023, we also observe similar results when applying a Fast Fourier Transform to MNIST data, finding the mean and variance of each frequency, and then apply an inverse FFT to get noise with similar frequency-statistics as the underlying dataset.

Whilst the above results are promising, a natural concern is the quality of solutions produced. In particular, if a significant fraction of networks trained using Modified Backpropagation have weights close to the null matrix $\mathbf{0}$ or the identity matrix $\mathbf{I}$ then the algorithm might not be practically useful. We refer the reader to Appendix D which shows that the norm of trained weight matrices in general is comparable to those found by Ordinary Backpropagation.

### 3.3. Relative Computational Cost

Theoretical analysis shows that Modified Backpropagation and Ordinary Backpropagation both require on the order of $\mathcal{O}(k)$ matrix multiplications for a $k$-layer MLP under minimal memoization assumptions. In Appendix E we provide the full analysis for this, as well as Figure 14 which shows empirically that the wall-clock running time of both algorithms is roughly the same for the networks B1-4.

Whilst we provide analysis for the MLP case here, we expect similar findings for other architectures. Practically, the major difference between Ordinary Backpropagation and Modified Backpropagation is the way gradients of the loss with respect to the output of the network is computed. In Modified Backpropagation, we require only forward passes of the network to calculate this quantity, whilst for Ordinary Backpropagation one must also find $\frac{\partial f_{\boldsymbol{\theta}}(\mathbf{y})}{\partial \mathbf{y}}$ due to the secondary application of $f$ in the loss function (Eq. 2). Thus, in implementations using memoization one should generally expect training time of both algorithms to differ only by a constant factor, whilst without memoization we generally expect Modified Backpropagation to have a computational advantage.

### 3.4. Application to Generative Networks

As mentioned, one of the motivating factors for actively enforcing idempotency during training is to apply it as a secondary optimization objective in conjunction with optimizing for a primary task. In this section we replicate the results of Shocher et al. 2023 as we train a U-net style DCGAN architecture (see Appendix F) on the MNIST and CelebA

datasets. Let $\mathcal{D}$ denote the distribution of dataset samples, while $\mathcal{D}'$ is a distribution from which noise is sampled. For MNIST we use $\mathcal{D}' = \mathcal{N}(0,1)$ whilst for CelebA we use a distribution of noise with similar frequency-statistics as the dataset, following Shocher et al. 2023. Let $\boldsymbol{\theta}'$ be a copy of the trainable weights $\boldsymbol{\theta}$ at each time step, where $\boldsymbol{\theta}'$ is detached from the computational graph. In this training scheme, the loss function being optimized is

$$
\begin{aligned}
\mathcal{L}(\boldsymbol{\theta}, \boldsymbol{\theta}') = {} & \lambda_r \mathcal{L}_{\text{rec}}(\boldsymbol{\theta}) \\
& + \lambda_i \mathcal{L}_{\text{idem}}(\boldsymbol{\theta}, \boldsymbol{\theta}') + \lambda_r \mathcal{L}_{\text{tight}}(\boldsymbol{\theta}, \boldsymbol{\theta}').
\end{aligned}
\tag{17}
$$

To see why employing two copies of the weights is useful, consider $(\mathbf{x}, \mathbf{y}^*) \sim \mathcal{D}$ and $\mathbf{z} \sim \mathcal{D}'$ and the individual loss components:

$$
\mathcal{L}_{\text{rec}}((\mathbf{x}, \mathbf{y}^*); \boldsymbol{\theta}) = \|\mathbf{y}^* - f_{\boldsymbol{\theta}}(\mathbf{x})\|_1
\tag{18}
$$

$$
\mathcal{L}_{\text{idem}}(\mathbf{z}; \boldsymbol{\theta}, \boldsymbol{\theta}') = \|f_{\boldsymbol{\theta}'}(f_{\boldsymbol{\theta}}(\mathbf{z})) - f_{\boldsymbol{\theta}}(\mathbf{z})\|_1
\tag{19}
$$

$$
\mathcal{L}_{\text{tight}}(\mathbf{z}; \boldsymbol{\theta}, \boldsymbol{\theta}') = -\|f_{\boldsymbol{\theta}}(f_{\boldsymbol{\theta}'}(\mathbf{z})) - f_{\boldsymbol{\theta}'}(\mathbf{z})\|_1
\tag{20}
$$

For instance, the quantity $\frac{\partial \mathcal{L}_{\text{idem}}(\mathbf{z}; \boldsymbol{\theta}, \boldsymbol{\theta}')}{\partial \boldsymbol{\theta}}$ is only affected by the inner application of $f$ above due to $\boldsymbol{\theta}'$ being detached from the computational graph. The relationship between loss components $\mathcal{L}_{\text{idem}}$ and $\mathcal{L}_{\text{tight}}$ is adversarial in nature.

The major difference in this work from Shocher et al. 2023 is that we use Modified Backpropagation for implementing both $\mathcal{L}_{\text{idem}}$ and $\mathcal{L}_{\text{tight}}$. As such, we *evaluate* both loss components as the Mean Squared Error (MSE) instead of the $L_1$ loss, and we use the training scheme in Section 2.3 to perform the backwards pass (see also Appendix F). We use the same implementation for $\mathcal{L}_{\text{rec}}$ as above.

We have successfully replicated several results of Shocher et al. 2023 under this training scheme. In particular, Figure 8 shows qualitative examples of noise drawn from $\mathcal{D}'$ being mapped to images resembling samples from the MNIST and CelebA datasets. While outputs remain largely similar between the first and second application of the network, in some cases we do also observe the same "self-correction" behaviour after the second application as observed by Shocher et al. 2023, with some small defects in background, hairstyle and facial features being corrected. Figure 10 gives further evidence for this, as we demonstrate the ability to recover original dataset images after various degradations have been applied, such as noise, greyscale filters, and Gaussian blur (see Appendix F for details).

In Figure 9 we visualize the effect of applying the trained network to noise linearly interpolated between two clear MNIST samples $\mathbf{A}, \mathbf{B} \in \mathbb{R}^{28 \times 28}$. We again observe the secondary application of the network "cleaning up" images. For more uncurated examples of generated images, see Appendix G.

We note that qualitative results in this training scheme for both Ordinary Backpropagation (as applied in Shocher et al.

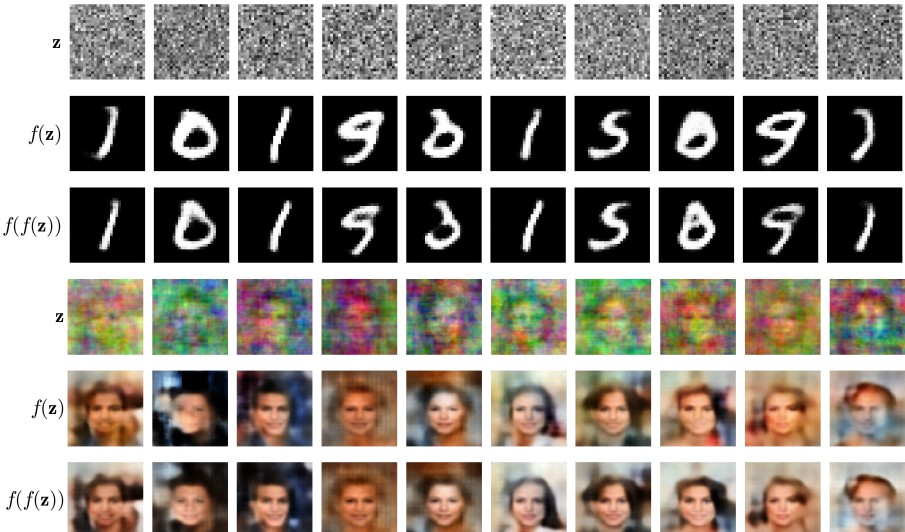

*Figure 8.* Uncurated generations of the U-net style DCGAN model trained on MNIST and CelebA with Modified Backpropagation for optimizing idempotent and tightness losses. Rows **z** denote samples drawn from $\mathcal{D}'$ whilst the second and third rows represent first and second application of the network to these samples, respectively.

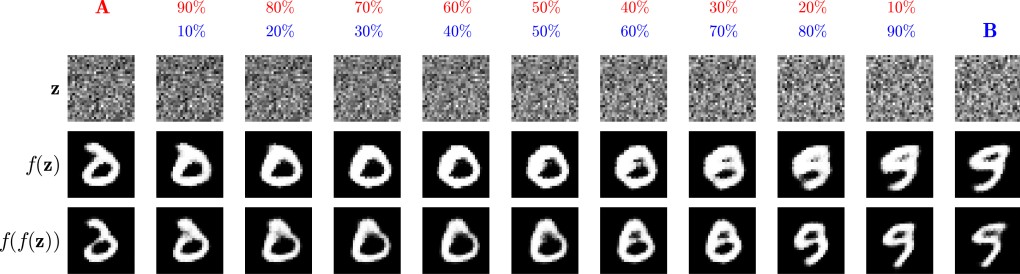

*Figure 9.* Latent space manipulation for MNIST with the model used in Figure 8. Samples **A** and **B** are selected randomly while remaining samples are linear combinations of these. We give the first and second application of the model.

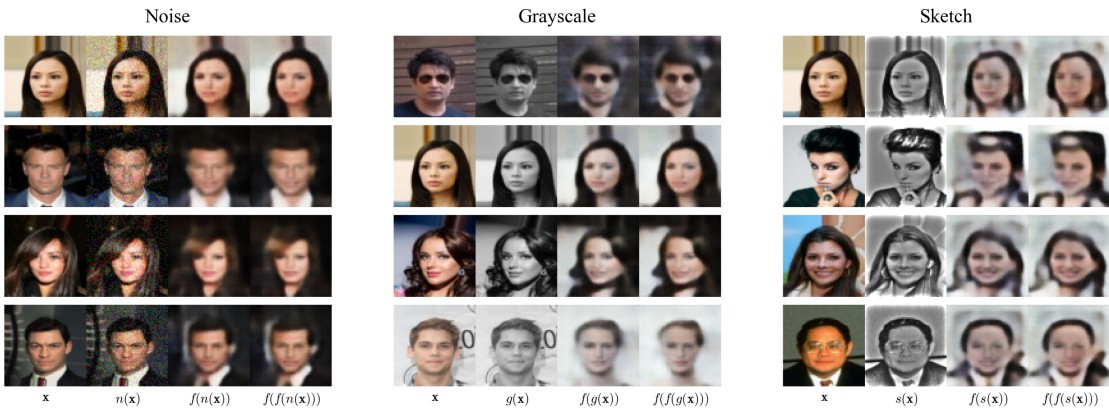

*Figure 10.* Out-of-distribution mappings CelebA with the model used in Figure 8. To images in the dataset we apply transformations as in Shocher et al. 2023 and plot the first and second application of the model. Characteristic to our method the images appear "blurry" but the model recovers the original image relatively well.

2023) and Modified Backpropagation are heavily sensitive to the hyperparameters $\lambda$ chosen in Eq. (17). Whilst results for MNIST are comparable to those obtained by Shocher et al. 2023, our results on CelebA are less competitive with their work and the state-of-the-art. Nevertheless, with fine-tuned hyperparameters we believe our results could be improved significantly and potentially be competitive with state-of-the-art models.

Although the aim of our results is to demonstrate practical applications of Modified Backpropagation, we believe the results could be easily replicated with other datasets, such as Cifar10, or similar. Additionally, further work is required to fully ascertain the practical benefit of Modified Backpropagation as a secondary optimization procedure in general. In particular, further experiments with other data modalities, primary optimization objectives, and datasets is needed to understand the wider applicability of the results of Sections 3.1 and 3.2.

## 4. Related Work

### 4.1. Algebraic Properties in Neural Networks

There has been significant work in actively enforcing algebraic properties in weights of neural networks for a variety of reasons. For instance, in Mikolov et al. 2015; Le et al. 2015 it was found that enforcing *part* of the weights of an RNN to remain close to the identity throughout training can cause the network to capture more long-term information, yielding performance close to LSTMs for the same natural language modelling tasks. Arjovsky et al. 2016 have also found that RNNs can overcome the exploding/vanishing gradient problem if weight matrices are actively enforced to be approximately unitary throughout training. Similarly, Saxe et al. 2014; Jing et al. 2017; Kiani et al. 2022 all explore unitary RNNs in the same vein. Lastly, Ardizzone et al. 2019 demonstrate that under a specialized training scheme, inverse mappings of MLPs can be found that can recover correlations in the parameter space.

These methods generally attempt to *mitigate undesirable effects* inherent to particular architectures or training schemes, whereas the work we present intends to *characterize behaviour* that the network should exhibit. As such, applications of the work of Shocher et al. 2023 is closest to ours, hence we focus on this in Section 3.4. Nevertheless, the prolific use of orthogonal/unitary weights in the literature invites future work into adapting Modified Backpropagation to enforce orthogonality $\left( (\mathbf{K}')^T(\mathbf{K}') = (\mathbf{K}')(\mathbf{K}')^T = \mathbf{I} \right)$ as opposed to idempotency $\left( (\mathbf{K}')^2 - (\mathbf{K}') = \mathbf{0} \right)$.

### 4.2. Alternatives to Gradients

There is a large body of work focusing on approximating gradients. In Spall 1999, a stochastic variant of finite dif-ferences called SPSA is suggested with competitive performance at lower relative computational cost. Bandler et al. 1988; Do & Reynolds 2013; Scheinberg 2022 all further explore this and similar approaches based on finite differences. Nevertheless, a core idea of Modified Backpropagation is the direct substitution of the gradient of $\mathcal{L}_{\text{idem}}$ with the quantity in Eq. (15). Although the exact relationship between this quantity and the gradients produced by Ordinary Backpropagation is still unclear, it is certainly not a trivial approximation as evidenced in Section 3.1.

## 5. Conclusion

In this work, we have given motivation for actively enforcing idempotency in arbitrary neural networks used in data transformation. The central idea presented is the idempotent corrector $g(\mathbf{K}) = 3\mathbf{K}^3 - 2\mathbf{K}^2$, which has been derived by solving a system of polynomial equations such that $\mathbf{K}' = g(\mathbf{K})$ for a near-idempotent matrix $\mathbf{K}$ and a perfectly idempotent $\mathbf{K}'$. We have also expanded the idea to a training scheme for arbitrary neural networks via a modification to the canonical backpropagation algorithm, termed Modified Backpropagation. Experimental results have shown that optimizer trajectories generally differ from those of Ordinary Backpropagation across a wide variety of MLP network configurations. Furthermore, we showed that Modified Backpropagation outperforms Ordinary Backpropagation in finding neural networks with lower idempotent error by up to an order of magnitude. Lastly, we demonstrate that Modified Backpropagation can be used alongside Ordinary Backpropagation to train generative models on MNIST and CelebA, following the example of Shocher et al. 2023.

We believe the work presented here suggests that alternative methods to gradient-based optimization in neural networks are practically viable, and we hope that future work will explore other applications of the central ideas presented here.

## Acknowledgements

The authors thank the anonymous reviewers for their helpful comments and suggestions.

## Impact Statement

This paper presents work whose goal is to advance the field of Machine Learning. There are many potential societal consequences of our work, none which we feel must be specifically highlighted here.

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

## A. Solutions to the Ansatz

We give a detailed description of how to derive idempotent correctors for matrices $\mathbf{K}$ that are near-idempotent to order $n$ with a fixed dimension $j$. For a given $j$, find a mapping $g$ making the input idempotent:

1. Assume $\mathbf{K} = \mathbf{P} + \mathbf{D}$.

2. Assume that $\mathbf{D}^2 \approx \mathbf{0}$, $\mathbf{P}^2 = \mathbf{P}$, and $\mathbf{XDYDZ} \approx \mathbf{0}$ for all $\mathbf{X}, \mathbf{Y}, \mathbf{Z}$.

3. Expand the expression $(\mathbf{K}')^2 - \mathbf{K}' = 0$.

   - $j = 2$ gives **34 terms**
   - $j = 3$ gives **154 terms**
   - $j = 6$ gives **10,794 terms**

4. Apply assumptions from 2 recursively.

   - $j = 2$ reduces 34 to **16 terms**
   - $j = 3$ reduces 154 to **32 terms**
   - $j = 6$ reduces 10,794 to **104 terms**

5. Collect coefficients for $\mathbf{D}$, $\mathbf{P}$, $\mathbf{DP}$, $\mathbf{PD}$, and $\mathbf{PDP}$ (no other exist).

6. Create a set of equations from coefficients and solve as a set of polynomial equations.

For $n = 1$ we give the first $j = 1, \ldots, 8$ idempotent correctors which have been found using the above recipe and the "Reduce" method in Mathematica:

- For $j = 1, 2$, there are no solutions.

- For $j = 3$, there is one solution (found in $\sim 130$ ms):

$$g(\mathbf{K}) = 3\mathbf{K}^2 - 2\mathbf{K}^3 \tag{21}$$

- For $j = 4$, there is one solution (found in $\sim 611$ ms):

$$g(\mathbf{K}) = \alpha_2\mathbf{K}^2 + (4 - 2\alpha_2)\mathbf{K}^3 + (\alpha_2 - 3)\mathbf{K}^4 \tag{22}$$

- For $j = 5$, there is one solution (found in $\sim 3.8$ s):

$$g(\mathbf{K}) = \alpha_2\mathbf{K}^2 + \alpha_3\mathbf{K}^3 + (5 - 3\alpha_2 - 2\alpha_3)\mathbf{K}^4 + (2\alpha_2 + \alpha_3 - 4)\mathbf{K}^5 \tag{23}$$

- For $j = 6$, there is one solution (found in $\sim 24$ s):

$$g(\mathbf{K}) = \alpha_2\mathbf{K}^2 + \alpha_3\mathbf{K}^3 + \alpha_4\mathbf{K}^4 + (6 - 4\alpha_2 - 3\alpha_3 - 2\alpha_4)\mathbf{K}^5 + (3\alpha_2 + 2\alpha_3 + \alpha_4 - 5)\mathbf{K}^6 \tag{24}$$

- For $j = 7$, there is one solution (found in $\sim 150$ s):

$$\begin{aligned} g(\mathbf{K}) = \alpha_2\mathbf{K}^2 + \alpha_3\mathbf{K}^3 + \alpha_4\mathbf{K}^4 + \alpha_5\mathbf{K}^5 + (7 - 5\alpha_2 - 4\alpha_3 - 3\alpha_4 - 2\alpha_5)\mathbf{K}^6 \\ + (4\alpha_2 + 3\alpha_3 + 2\alpha_4 + \alpha_5 - 6)\mathbf{K}^7 \end{aligned} \tag{25}$$

- For $j = 8$, there is one solution (found in $\sim 920$ s):

$$\begin{aligned} g(\mathbf{K}) = \alpha_2\mathbf{K}^2 + \alpha_3\mathbf{K}^3 + \alpha_4\mathbf{K}^4 + \alpha_5\mathbf{K}^5 + \alpha_6\mathbf{K}^6 + (8 - 6\alpha_2 - 5\alpha_3 - 4\alpha_4 - 3\alpha_5 - 2\alpha_6)\mathbf{K}^7 \\ + (5\alpha_2 + 4\alpha_3 + 3\alpha_4 + 2\alpha_5 + \alpha_6 - 7)\mathbf{K}^8 \end{aligned} \tag{26}$$

## B. Jordan Normal Form Analysis

For a matrix $\mathbf{K} \in \mathbb{R}^{n \times n}$, let $\alpha(\lambda)$ denote the algebraic multiplicity of eigenvalue $\lambda$ and let $\gamma(\lambda)$ denote the geometric multiplicity of $\lambda$. Given the equation

$$\mathbf{K} = 3\mathbf{K}^2 - 2\mathbf{K}^3, \tag{27}$$

we can substitute $\mathbf{K} = \mathbf{PJP}^{-1}$,

$$\mathbf{PJP}^{-1} = 3(\mathbf{PJP}^{-1})^2 - 2(\mathbf{PJP}^{-1})^3, \tag{28}$$

which can be simplified to

$$\mathbf{PJP}^{-1} = 3\mathbf{PJ}^2\mathbf{P}^{-1} - 2\mathbf{PJ}^3\mathbf{P}^{-1}. \tag{29}$$

Since $\mathbf{P}$ is invertible, this is equivalent to

$$\mathbf{J} = 3\mathbf{J}^2 - 2\mathbf{J}^3. \tag{30}$$

Thus for an arbitrary $\mathbf{K}$, solving Eq. (27) equates to solving Eq. (30).

Since $\mathbf{J}$ is block-diagonal, the equation in (30) can be broken down into smaller equations for each block, $\mathbf{J}_\lambda = 3\mathbf{J}_\lambda^2 - 2\mathbf{J}_\lambda^3$, which we can write as a system of equations:

$$\begin{pmatrix} \lambda & 1 & 0 & \dots & 0 \\ 0 & \lambda & 1 & \dots & 0 \\ 0 & 0 & \lambda & \dots & 0 \\ \vdots & \vdots & \vdots & \ddots & \vdots \\ 0 & 0 & 0 & \dots & \lambda \end{pmatrix} = 3 \begin{pmatrix} \lambda^2 & 2\lambda & 1 & \dots & 0 \\ 0 & \lambda^2 & 2\lambda & \dots & 0 \\ 0 & 0 & \lambda^2 & \dots & 0 \\ \vdots & \vdots & \vdots & \ddots & \vdots \\ 0 & 0 & 0 & \dots & \lambda^2 \end{pmatrix} - 2 \begin{pmatrix} \lambda^3 & 3\lambda^2 & 3\lambda & 1 & \dots & 0 \\ 0 & \lambda^3 & 3\lambda^2 & 3\lambda & \dots & 0 \\ 0 & 0 & \lambda^3 & 3\lambda^2 & \dots & 0 \\ \vdots & \vdots & \vdots & \vdots & \ddots & \vdots \\ 0 & 0 & 0 & 0 & \dots & \lambda^3 \end{pmatrix} \tag{31}$$

Thus, for a $(k \times k)$ Jordan block we have up to four equations:

$$\lambda = 3\lambda^2 - 2\lambda^3 \tag{32}$$
$$1 = 3(2\lambda) - 2(3\lambda^2) = 6\lambda - 6\lambda^2 \qquad \text{Only when } k \geq 2. \tag{33}$$
$$0 = 3(1) - 2(3\lambda) = 3 - 6\lambda \qquad \text{Only when } k \geq 3. \tag{34}$$
$$0 = 3(0) - 2(1) = 0 - 2 \qquad \text{Only when } k \geq 4. \tag{35}$$

There are never more equations than this, since all other entries in a Jordan block must be 0.

Since Eq. (35) is a contradiction, we can have no solution which solves all Eqs. (32), (33), (34), and (35). Note also that there exists no solutions satisfying Eqs. (32), (33), and (34), nor do any solutions exist for both Eqs. (32) and (33). The following are solutions which satisfy only Eq. (32):

$$\lambda = \{0, 0.5, 1\} \tag{36}$$

The only situation where Eqs. (33), (34), and (35) do not arise is when $\alpha(\lambda) = \gamma(\lambda)$, which is precisely the case when $\mathbf{K}$ is diagonalizable. Therefore, any $\mathbf{K}$ which is a solution to (27) must have a Jordan normal form which is a diagonal matrix:

$$\mathbf{J} = \begin{pmatrix} \lambda_1 & 0 & 0 & \dots & 0 \\ 0 & \lambda_2 & 0 & \dots & 0 \\ \vdots & \vdots & \vdots & \ddots & \vdots \\ 0 & 0 & 0 & \dots & \lambda_3 \end{pmatrix} \tag{37}$$

# C. Automatic Differentiation Rule

To implement Modified Backpropagation in PyTorch we make use of custom autograd functions. The mathematical description of the loss function for idempotency is:

$$\mathcal{L}_{\text{idem}}(\mathbf{y}) = \frac{1}{n} \sum_{i=1}^{n} \left( f_{\boldsymbol{\theta}}(\mathbf{y}) - \mathbf{y} \right)^2. \tag{38}$$

To practically implement the function, we note that PyTorch optimizers reduce $-\mathcal{L}$ for some loss function $\mathcal{L}$, but as per Section 2, our method reduces idempotent error when the loss function above is *not* negatively signed. This leads to the implementation in the `E.backward` function in algorithm 1.

---

**Algorithm 1** Modified Backpropagation PyTorch rule.

---

```python
class E(torch.autograd.Function):
    @staticmethod
    def loss_fn(y, net):
        loss = torch.mean((net(y) - y) ** 2)
        return loss

    @staticmethod
    def forward(ctx, y, net):
        ctx.save_for_backward(y)
        ctx.net = net

        return E.loss_fn(y, net)

    @staticmethod
    def backward(ctx, grad_output):
        y, = ctx.saved_tensors
        net = ctx.net

        y2 = net(y)
        y3 = net(y2)
        e = 3*y2 - 2*y3 - y
        grads = -e / e.shape[0]
        return grads * grad_output, None

class ELoss(torch.nn.Module):
    def __init__(self, net, mode):
        super(ELoss, self).__init__()
        self.net = net
        self.mode = mode

    def forward(self, y):
        return E.apply(y, self.net)
```

---

We give the computational graph constructed by PyTorch for calculating gradients in Modified Backpropagation and Ordinary Backpropagation. The graphs are constructed using the same single-layer neural network without biases and no activation function. We therefore have $f_{\boldsymbol{\theta}}(\mathbf{x}) = \mathbf{W}\mathbf{x}$. **PyTorch, however, represents the input $\mathbf{x}$ transposed, hence the following use $f_{\boldsymbol{\theta}}(\mathbf{x}) = \mathbf{W}\mathbf{x}^T = \mathbf{x}\mathbf{W}^T$ as the definition.** We show the graphs describing the computation of gradients for $\mathbf{W}$ for a single optimization step.

In these graphs, the weight matrix $\mathbf{W}$ is the blue box, and the green box at the bottom of the graph is the gradient matrix $\frac{\partial \mathcal{L}_{\text{idem}}(f_{\boldsymbol{\theta}}(\mathbf{x}))}{\partial \mathbf{W}}$. Yellow boxes of size $3 \times 5$ denote the input $\mathbf{x}$ consisting of 3 samples with 5 features each.

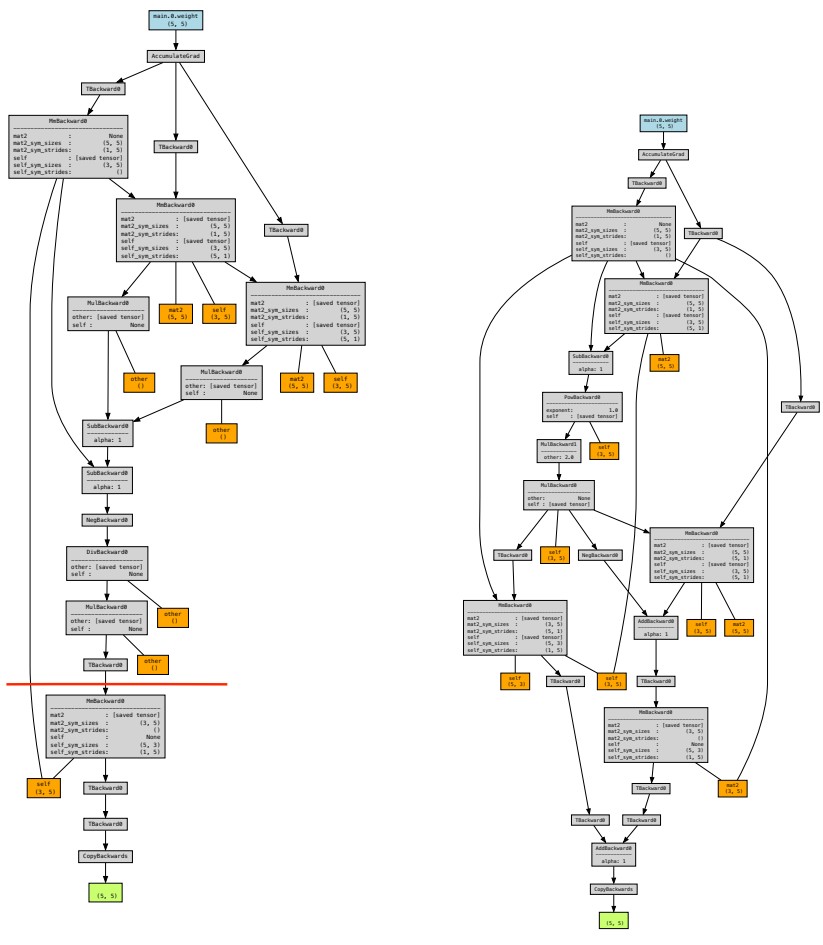

(a) **Modified Backpropagation**      (b) **Ordinary Backpropagation**

*Figure 11.* PyTorch computational graphs for gradient calculation.

The red line in Figure 11(a) denotes the intuitive split by the chain rule. Computation above the line corresponds to $\frac{\partial \mathcal{L}_{\text{idem}}(\mathbf{y})}{\partial \mathbf{y}}$ and computation below the line corresponds to $\frac{\partial f_{\boldsymbol{\theta}}(\mathbf{x})}{\partial \mathbf{W}}$, which altogether gives $\frac{\partial \mathcal{L}_{\text{idem}}(f_{\boldsymbol{\theta}}(\mathbf{x}))}{\partial \mathbf{W}}$. Chasing through the graph shows that it indeed does use exactly the algorithm outlined above.

The graph in Figure 11(b) shows a computation of $\frac{\partial \mathcal{L}_{\text{idem}}(f_{\boldsymbol{\theta}}(\mathbf{x}))}{\partial \mathbf{W}}$ without using E.backward. Although the graph is less obvious, chasing the graph shows that it computes the following quantity:

$$(2(\mathbf{x}\mathbf{W}^T\mathbf{W}^T - \mathbf{x}\mathbf{W}^T) \cdot \mathbf{x})^T \mathbf{x}\mathbf{W}^T + ((2(\mathbf{x}\mathbf{W}^T\mathbf{W}^T - \mathbf{x}\mathbf{W}^T) \cdot \mathbf{x})\mathbf{W} - 2(\mathbf{x}\mathbf{W}^T\mathbf{W}^T - \mathbf{x}\mathbf{W}^T) \cdot \mathbf{x})^T \mathbf{x}.$$

This is equivalent to the ordinary analytical solution to $\frac{\partial \mathcal{L}_{\text{idem}}(f_{\boldsymbol{\theta}}(\mathbf{x}))}{\partial \mathbf{W}}$ for this particular network.

# D. Idempotent Error and Norm for Test Networks

For each of the test networks outlined in Table 1 we report the distribution of absolute idempotent error and norm of the resulting mapping in Figures 12 and 13. The learning rates used for training of each network is chosen to be the value in Figure 6 with the lowest idempotent error for each algorithm and network configuration.

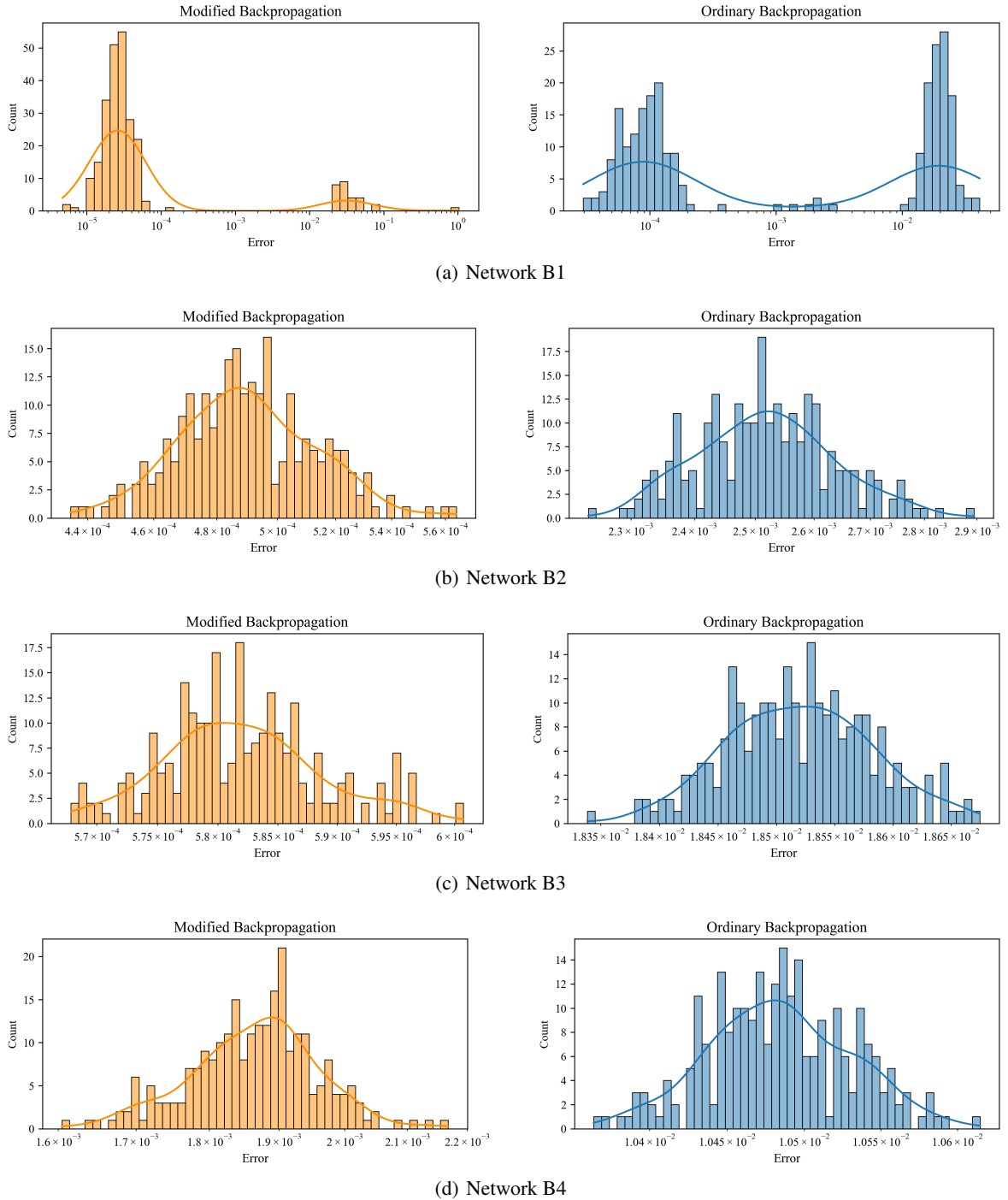

*Figure 12.* For each network we report the distribution of **absolute idempotent error** after 2500 epochs of training 250 randomly initialized networks. Note that distributions are narrow for each algorithm and often separated by more than an order of magnitude.

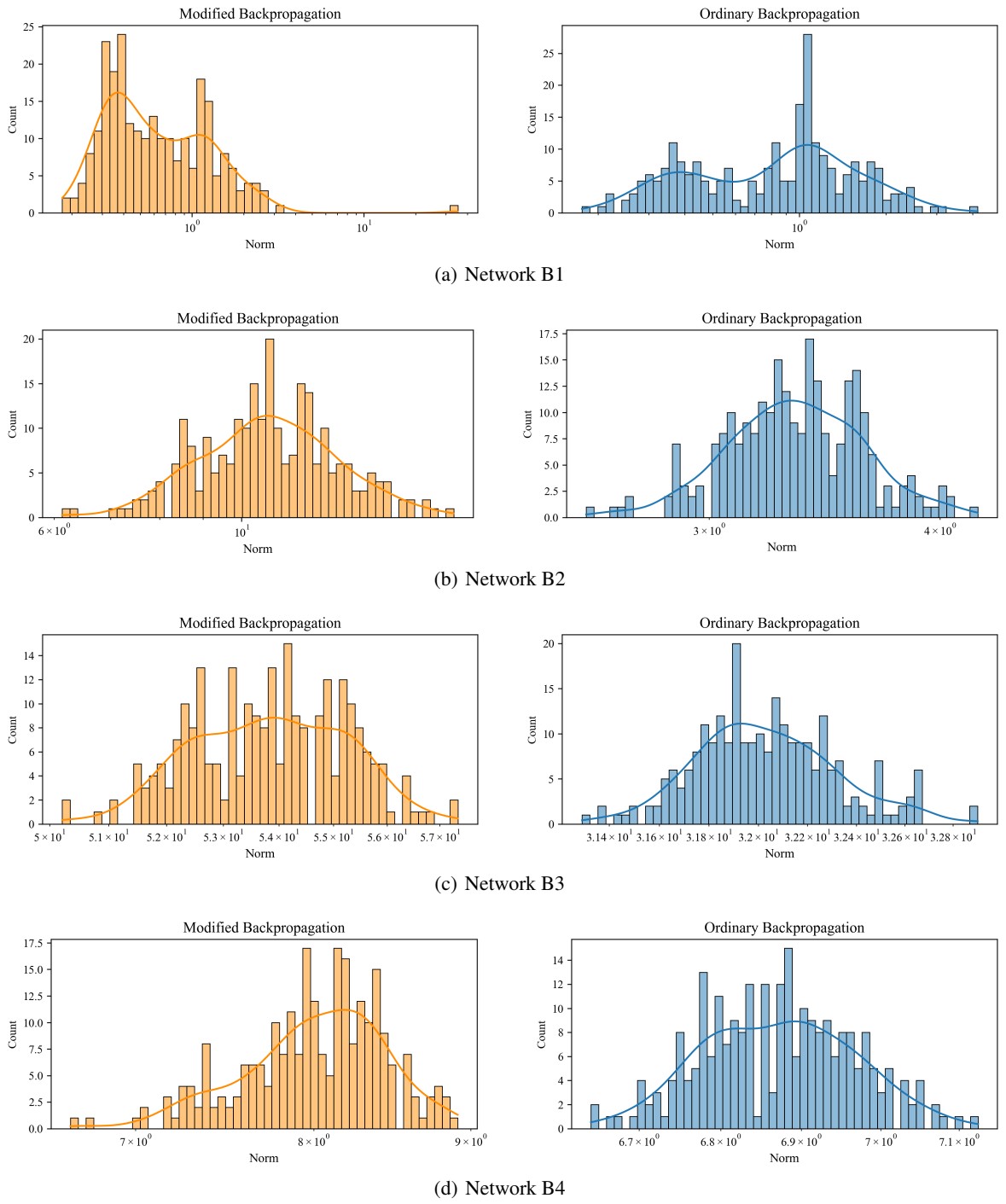

*Figure 13.* For each network we report the distribution of **norms** after 2500 epochs of training 250 randomly initialized networks. Note that distributions for either method are not centred around zero, indicating that the trained network is a non-trivial idempotent mapping. To calculate the norm we pass each canonical basis vector of $\mathbb{R}^n$ as input by concatenating them into the identity matrix and report here the Frobenius norm of the output matrix.

## E. Relative Computational Cost

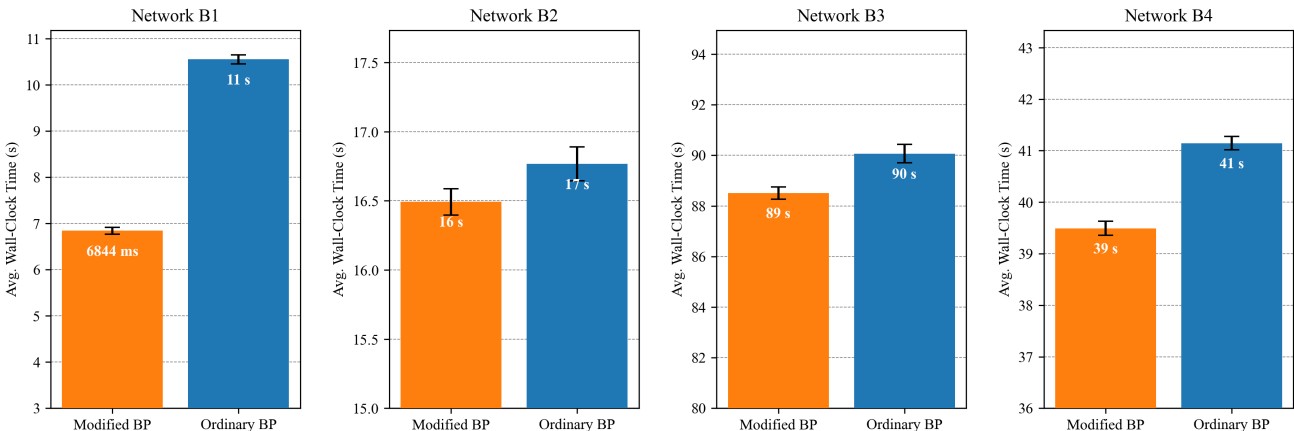

*Figure 14.* Computational cost for networks B1-B4 in Table 1. Plot of average wall-clock running time consumed per training epoch across 250 repetitions of each algorithm, with 98% confidence intervals. Across all configurations, both algorithms take approximately the same amount of time, with a slight advantage to Modified Backpropagation.

The practical findings of Figure 14 are supported by a theoretical argument which we give in detail here. Concretely, we consider the growth in the number of matrix multiplications required as the number of layers increases in an ordinary MLP. Defining $\mathbf{y} = f_{\boldsymbol{\theta}}(\mathbf{x})$, the loss function in Eq. 2 can be written as $\mathcal{L}'_{\text{idem}}(\mathbf{y}) = \frac{1}{m} \sum (f_{\boldsymbol{\theta}}(\mathbf{y}) - \mathbf{y})^2$. By the chain rule $\frac{\partial \mathcal{L}'_{\text{idem}}}{\partial \mathbf{W}} = \frac{\partial \mathcal{L}'_{\text{idem}}}{\partial \mathbf{y}} \frac{\partial \mathbf{y}}{\partial \mathbf{W}}$. Computing $\frac{\partial \mathbf{y}}{\partial \mathbf{W}}$ using backpropagation will generally use $\mathcal{O}(k)$ matrix multiplications for a $k$-layer MLP. This quantity is computed in the same way for both Ordinary Backpropagation and Modified Backpropagation.

For Ordinary Backpropagation, the quantity $\frac{\partial \mathcal{L}'_{\text{idem}}}{\partial \mathbf{y}}$ can also be unfolded via the chain rule and its evaluation requires computing $\mathbf{y} = f_{\boldsymbol{\theta}}(\mathbf{x})$, $f_{\boldsymbol{\theta}}(\mathbf{y})$, as well as $\frac{\partial f_{\boldsymbol{\theta}}(\mathbf{y})}{\partial \mathbf{y}}$, which each can be computed in $\mathcal{O}(k)$ time. Thus, assuming memoization is used (as is the case by design in most automatic differentiation frameworks, like PyTorch), Ordinary Backpropagation can compute $\frac{\partial \mathcal{L}'_{\text{idem}}}{\partial \mathbf{W}}$ in $\mathcal{O}(k)$ time.

For Modified Backpropagation we compute $\frac{\partial \mathcal{L}'_{\text{idem}}}{\partial \mathbf{y}} = 3f_{\boldsymbol{\theta}}(\mathbf{y}) - 2f_{\boldsymbol{\theta}}(f_{\boldsymbol{\theta}}(\mathbf{y})) - \mathbf{y}$. Again, assuming memoization, each of these terms take $\mathcal{O}(k)$ time to evaluate, so Modified Backpropagation also runs in $\mathcal{O}(k)$ time.

Therefore, both Ordinary Backpropagation and Modified Backpropagation should require the same number of matrix multiplications under realistic assumptions.

# F. Training Scheme for Generative Models

We follow the example of Shocher et al. 2023 and train a U-Net style DCGAN architecture on the MNIST dataset. Let $\mathcal{Y}$ denote the latent space and $\mathcal{X}$ denote the sample space. Then, letting $G : \mathcal{Y} \to \mathcal{X}$ and $D : \mathcal{X} \to \mathcal{Y}$ be the Generator and Discriminator of the network respectively, we apply the network $G(D(\mathbf{x})) = \mathbf{y}$.

Table 2 explicitly shows the architecture we use, and Table 3 shows the hyperparameters chosen for training. The core difference between our configuration and that of Shocher et al. 2023 is the use of dropout layers at some layers.

*Table 2.* U-Net style DCGAN Network architecture.

| | Layer | Size | Stride | Padding | Features | Dropout? | BN? | Activation Func |
|---|---|---|---|---|---|---|---|---|
| Discriminator | Conv2D | $4 \times 4$ | 2 | 1 | 64 | ✓ | ✗ | LeakyReLU(0.2) |
| | Conv2D | $4 \times 4$ | 2 | 1 | 128 | ✓ | ✓ | LeakyReLU(0.2) |
| | Conv2D | $3 \times 3$ | 1 | 0 | 256 | ✓ | ✓ | LeakyReLU(0.2) |
| | Conv2D | $3 \times 3$ | 1 | 0 | 512 | ✓ | ✓ | LeakyReLU(0.2) |
| | Conv2D | $3 \times 3$ | 1 | 0 | 512 | ✗ | ✗ | None |
| Generator | ConvTranspose2D | $3 \times 3$ | 1 | 0 | 256 | ✓ | ✓ | ReLU |
| | ConvTranspose2D | $3 \times 3$ | 1 | 0 | 128 | ✓ | ✓ | ReLU |
| | ConvTranspose2D | $3 \times 3$ | 1 | 0 | 64 | ✓ | ✓ | ReLU |
| | ConvTranspose2D | $4 \times 4$ | 2 | 1 | 32 | ✓ | ✓ | ReLU |
| | ConvTranspose2D | $4 \times 4$ | 2 | 1 | 1 | ✗ | ✗ | Tanh |

*Table 3.* Training parameters.

| Parameter | Value |
|---|---|
| Reconstruction loss metric | $L_1 = \|\mathbf{y}^* - \mathbf{y}_{\text{pred}}\|_1$ |
| Loss terms weighting (MNIST) | $\lambda_r = 20, \lambda_i = 0.1, \lambda_r = 0.1$ |
| Loss terms weighting (CelebA) | $\lambda_r = 20, \lambda_i = 0.006, \lambda_r = 0.02$ |
| Optimizer | Adam(lr $= 1.0 \times 10^{-4}, \beta_1 = 0.5, \beta_2 = 0.999$) |
| Dropout probability | 0.05 |
| Batch Size | 512 |
| Epochs | 100 |
| Weight initialization | Default Kaiming Uniform initialization: $\mathcal{U}(-\sqrt{k}, \sqrt{k})$, for $k = \sqrt{1/n}$ for $n$ features. |

**Degradations applied for out-of-distribution mappings**

All degradations are implemented as in Shocher et al. 2023 and are restated here for clarity. Let $\mathbf{x}$ be an arbitrary sample from the dataset.

**Noise**: Let $n(\mathbf{x}) = \mathbf{x} + \mathbf{n}$, where $\mathbf{n}$ is sampled from $\mathcal{N}(0, 0.15)$.

**Greyscale**: Let $g(\mathbf{x})$ be the function that averages each pixel value across channels and returns the result.

**Gaussian Blur**: Let $s(\mathbf{x}) = \frac{g(\mathbf{x}+1)}{\text{gaussian\_blur}(g(\mathbf{x}+1),\, 21)+10^{-10}} - 1$, where gaussian_blur($\mathbf{z}$, 21) applies a Gaussian Blur with kernel size 21 to $\mathbf{z}$.

# G. Generative Model Sample Images

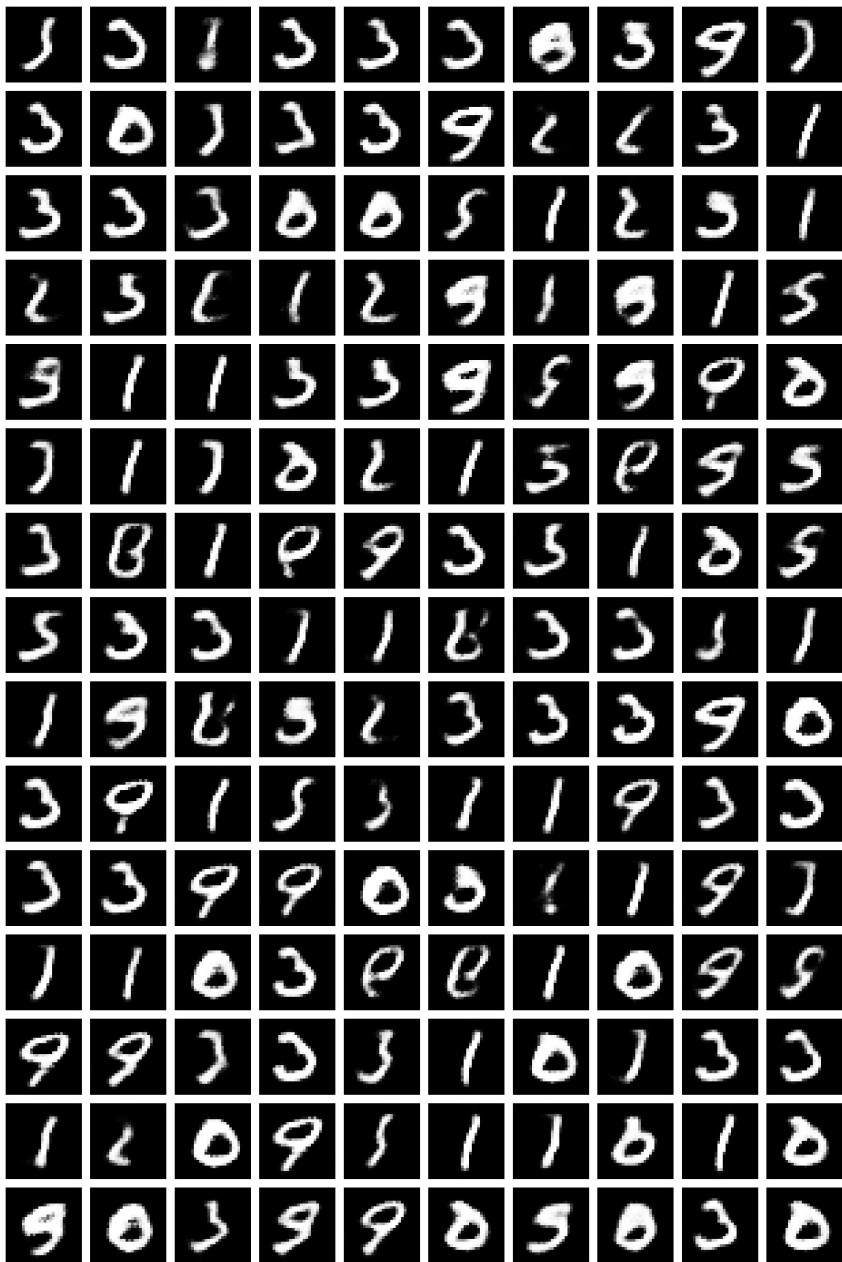

*Figure 15.* Uncurated generations of U-net style DCGAN model trained on MNIST with Modified Backpropagation for optimizing idempotent and tightness losses. All samples are drawn from a random distribution with mean 0 and variance 1, and the result of applying the network is shown.

## H. Optimizer Trajectory Plots

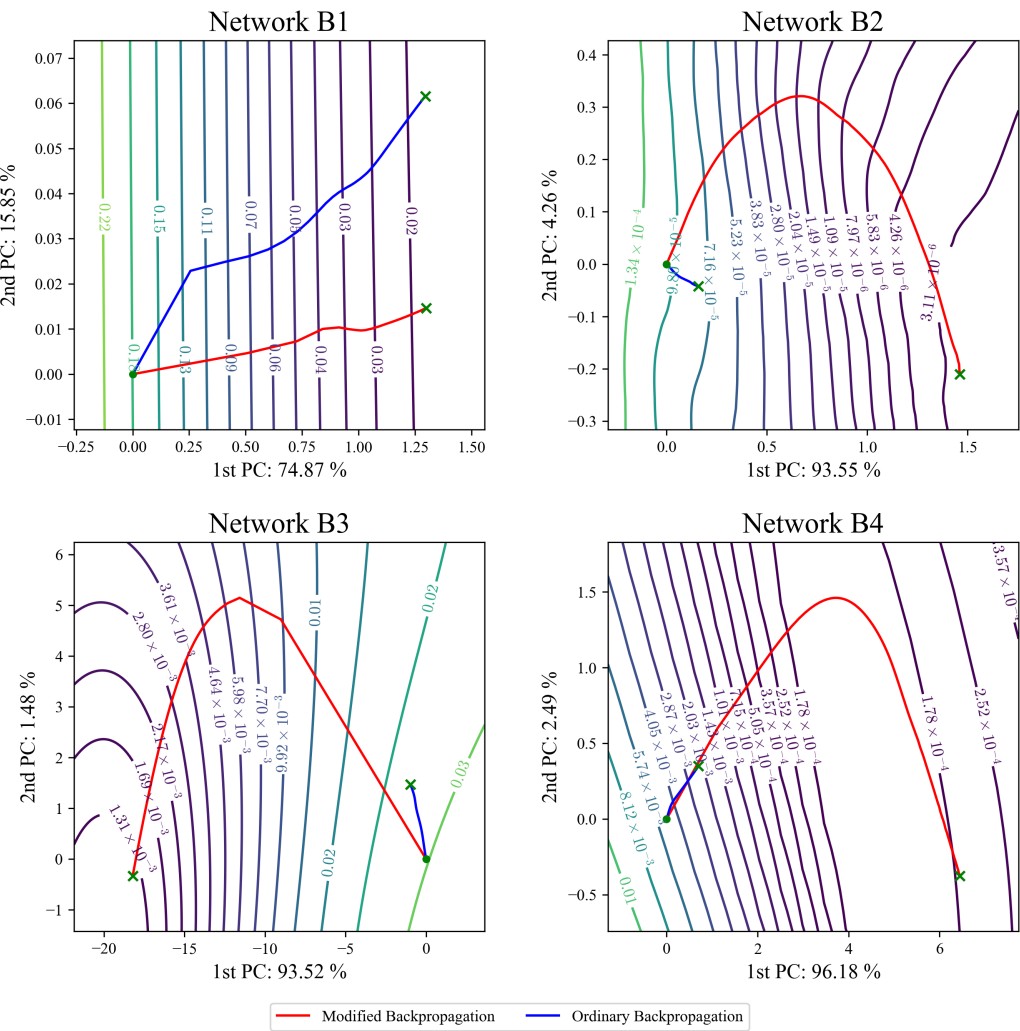

*Figure 16.* Representative projections of the optimizer trajectories over 2500 epochs of either algorithm on each of the B1-B4 models at optimal learning rates (Figure 6). Total variance captured is $> 90\%$ with cosine similarity of PC1 and PC2 less than $1.0 \times 10^{-6}$ across all plots. Optimizer trajectory of Modified Backpropagation often deviates significantly from Ordinary Backpropagation, but may sometimes overlap (*e.g.*, B4).

