# OpenReview forum: "Enforcing Idempotency in Neural Networks"
_ICML.cc/2025/Conference — ICML 2025 poster_

### Official Review · Reviewer_pCL4 · 2025-03-11

**Overall Recommendation:** 4

**Summary:**

Idempotent Generative Networks (IGN) require an operator $f$ to satisfy $f = f \circ f$. This paper addresses such idempotency by
 analysis from perturbation theory, identifying the polynomial $3K^2 - 2K^3$ as one that projects matrices to the idempotent matricse manifold. The authors then adapt this approach to the non-linear case. Specifically, they override the gradient of $||f(f(x))-f(x)||$ with $3f(y) - 2f(f(y)) - y$, thus simplifying backprop. Experiments on MLPs and on MNIST with a U-net style DCGAN show reduced idempotent error.

I am listing strengths, weaknesses, and questions throughout these review fields, marked as **(S), (W), (Q)**.


## update after rebuttal
As mentioned in my comment below, after some of my concerns have been addressed I would like to maintain my score and to state that this paper is safely in the "4" zone.
Thanks again to the authors.

**Claims And Evidence:**

**(W1)** The paper claims the polynomial-based gradient step outperforms naive backprop in achieving $f(f(x)) \approx f(x)$, demonstrated with synthetic MLPs and a small U-net on MNIST. While the results are persuasive for those cases, there is no large-scale benchmark comparison.

**Essential References Not Discussed:**

None that are strictly missing, but further parallels to other “fixed-point” networks or to full second-order approaches could be informative.

**Experimental Designs Or Analyses:**

**(S3)** They present ablation across different MLP depths and widths, showing a consistent drop in idempotent error. On MNIST, they adapt IGN method with “tightness” and show visually plausible results.

**(W4)** The dataset variety is minimal (mostly random MLP inputs, plus MNIST). Additional experiments on complex images or tasks would strengthen the claims.

**Methods And Evaluation Criteria:**

**(S1)** The authors propose a new method—“modified backprop”—that replaces $\partial L/\partial y$ by $3f(y)-2f(f(y))-y$, avoiding the usual chain rule on $f\circ f$. This is simple to implement and apparently stable.

**(W2)** Evaluation focuses on how quickly $\|\,f(f(x)) - f(x)\|\,$ shrinks. There is no direct measure of runtime or memory cost versus naive double-backprop, which would have clarified efficiency gains.

**Other Comments Or Suggestions:**

I find this paper an important contribution, taking a step with a currently not very practical generative model, making it more feasible. I think this has much potential, having in mind the first generative diffusion models paper (Sohl-Dickstein 2015) that took some time to adapt until it became a major thing. I am also impressed with the Interdisciplinarity and the idea of bringing in perturbation theory to solve this problem. I find it novel, creative and out of the main stream works. However, the experimental side of this work is not strong enough. I would expect a step up in terms of scale, data, quality w.r.t. the previous IGN paper. I see the experimental and evaluation side as problematic. Nevertheless I think this is a definite accept.

**Other Strengths And Weaknesses:**

**(S6)** This is a very elegant idea and an impressive solution. At first glance it seems impossible that gradients taken only through one copy of the network yied optimization equivalent to taking the through recursion of two copies of the network. This is a solid mathematical solution.

**(S7)** What the authors propose provides stability and freedom for IGN. It allows architectures that couldn't work with the original IGN and makes training more stable. This is a step towards making IGN practical.

**(W5)** I would want to see projection of out-of-distribution data as shown in the original IGN paper, I think it is a key feature of IGN.

**(S8)** the convergence diagram at Fig.2 is eye opening and very insightful.

**Questions For Authors:**

**(Q1)** I was trying to interpret what is being optimized by the modified backprop step, and I speculate it can be seen as the idempotency of the Jacobian of $f$ at location $x$. Specifically, it is the distance between the Jacobian and the polynomial of that Jacobian $||(3J_\theta(x)^2 - 2J_\theta(x)^3 - J_\theta(x))x||^2$.
1. In the linear setting, where $f(x) = Kx$ is constant, directly enforcing $K = 3K^2 - 2K^3$ yields idempotency.
2. For the non-linear case, one could want each local Jacobian$J_\theta(x)$ to act similarly, but that Jacobian depends on $\theta$ and $x$ and is not fixed. So what one can do is train the network weights so that the Jacobian is close to the polynomial, providing the objective  $L = E_x[||(3J_\theta(x)^2 - 2J_\theta(x)^3 - J_\theta(x))x||^2] = E_x[||3f(y) - 2f(f(y)) - y||^2]$.
3. Taking the gradient of this loss with respect to $y$ in such a construction is proportional to $3f(y) - 2f(f(y)) - y$, which matches the polynomial update the authors propose in their modified back-propagation.
4. This equivalence suggests the network is being trained so that, at for all inputs (noise or images) in the training distribution, the first order approximation is close to its projection onto the manifold of idempotent matrices.

**Relation To Broader Scientific Literature:**

**(S5)** The paper references Shocher et al. (Idempotent Generative Networks) and classical perturbation theory for linear idempotent operators. This is interdisciplinary, connecting different fields and gets benefits. This topic is mostly self-contained so not too much context is needed, I think the authors describe the relation well

**Theoretical Claims:**

**(S2)** The linear case for $K$ is well understood: $3K^2 - 2K^3$ projects $K$ onto the set of idempotent operators. The paper extends this to non-linear networks via the gradient substitution. This is conceptually clean, though no convergence proof is provided for deep architectures.

**(W3)** The derivation relies on near-idempotency perturbation arguments but does not fully address whether one can recover from being far from idempotent.

---

> ### Author Rebuttal · Authors · 2025-03-31
>
> We thank the reviewer for their time and valuable feedback. We are pleased that the reviewer finds this to be an important contribution with much potential, and with a strongly interdisciplinary character. We appreciate the concerns raised, and we aim to address these below.
>
> **(W2) Runtime and memory cost.** Please see our response to reviewer "kYga" for a discussion of the runtime cost of Modified Backpropagation vs. Ordinary Backpropagation.
>
> **(W4 + W5) Dataset variety and out-of-distribution experiments.** Please see our response to reviewer "hBam" where we give indicative results on CelebA and replicate the out-of-distribution experiment of the IGN paper.
>
> **(W3) Reliance on near-idempotency.** Our illustration Figure 2 shows a relatively large domain of convergence around the fixed points for the linear case. In the generative setting, other elements of the loss function already help the network to learn an approximately idempotent behaviour, moving the model weights to a near-idempotent regime where our method can be most effective. Questions of convergence in the general non-linear case are fascinating to consider and we propose to investigate this in further work.
>
> **(Q1).** The reviewer's perspective on how our method extends the idempotence condition from the linear to the non-linear case is very well-stated, and agrees with our current understanding. We are grateful to the reviewer for setting this out, and if accepted, we propose to revise the start of Section 2.3 to include this perspective.
>
> When extending Modified Backpropagation to the non-linear case, we wish for the network to act in an idempotent way around inputs taken from the training distribution (and hope that enough such points yields idempotent behaviour for the rest of the distribution). Indeed, by treating the behaviour of the network around inputs as locally linear, Eq. 15 becomes the ``obvious'' choice for optimizing idempotency in the context of the method we discuss in Section 2 of our paper. We therefore agree with the reviewer's suggestion that setting $\frac{\partial{L}}{\partial{\mathbf{y}}} = 3f(\mathbf{y}) - 2f(f(\mathbf{y})) - \mathbf{y}$ in Modified Backpropagation effectively trains the first order approximation of the network to act as an idempotent mapping. In particular, what we are optimizing is the idempotence property of the Jacobian of $f$ around input points.

---

> > ### Comment · Reviewer_pCL4 · 2025-04-01
> >
> > I thank the authors for their response.
> >
> > Most of my concerns have been addressed:
> >
> > **(W4+W5)** The authors provided results on another, slightly bigger dataset CelebA and showed projection from OOD. While not perfect and I think inferior to original IGN results, it seems to be possible and the gap is some tuning and surface engineering.
> >
> > **(W3)** I'm glad the authors pointed me to Fig.2 as it does show the tendency for the the idempotency to facilitate long-term. Of course it is hard to determine in general, but seems to be reasonable.
> >
> > **(W2)** I appreciate the theoretical compute calculation and convinced that theoretically there should be the same runtime, with faster convergence which makes it more efficient. I would have been more convinced by empirical evaluation as there can be overheads.
> >
> > Before the rebuttal my opinion was that this paper is a definite accept, with meaningful scientific contribution and elegant interdisciplinarity. In their response, the authors showed that their method is indeed more efficient and capable of handling more data and projection task. It is a bit disappointing  that results-wise there is no step forward from the original IGN, but it seems like this kind of work helps getting there by allowing stability and more architectural freedom. Overall, my concerns were mostly addressed. I don't think this paper should be rated 5. However, I can state that as I see it now, this work is more safely located in the 4 rating than before. I think this paper needs to be accepted.
> >
> >
> > **(Q1)** Yes, you have my permission to add this analysis to the paper if you wish to do so.

---

### Official Review · Reviewer_hBam · 2025-03-14

**Overall Recommendation:** 3

**Summary:**

This paper introduces a novel approach for training idempotent neural networks.
Leveraging techniques from perturbation theory on idempotent matrices, the authors propose a new method for projecting matrices onto the idempotent manifold.
They further extend this approach to nonlinear neural networks.
Finally, the paper presents experiments, primarily on toy datasets and MNIST generation.

**Claims And Evidence:**

I find the claims to be convincing.
A key challenge arises when extending linear idempotency to the nonlinear case.
To address this in a practical and reasonable manner, the authors approximate the gradient of the loss as a local linear matrix and incorporate its direction into the gradient update to reduce idempotency error. I believe this approach is fairly reasonable.

**Essential References Not Discussed:**

.

**Experimental Designs Or Analyses:**

- The visualizations of optimizer trajectories (Figure 3), absolute cosine similarity (Figure 4), and gradient norms (Figure 5) demonstrate that idempotent guidance facilitates the training of IGN. These visualizaton (Figures 3–7) provide valuable insights into the properties emerging from this idempotency guidance, effectively illustrating the benefits of the modified backpropagation.

- A major limitation of this paper is that the experiments are restricted to toy datasets and MNIST. Could the authors extend the experiments to CelebA, following the experiment protocol in IGN?

**Methods And Evaluation Criteria:**

Yes. More discussions on "Experimental Designs Or Analyses" section.

**Other Comments Or Suggestions:**

.

**Other Strengths And Weaknesses:**

.

**Questions For Authors:**

Could the authors provide a similar analysis for the MNIST experiments as presented in Figures 3–7? It would be valuable to see whether the observed properties hold in a larger-scale setting.

**Relation To Broader Scientific Literature:**

.

**Theoretical Claims:**

Theory in idempotency seems to be correct.

---

> ### Author Rebuttal · Authors · 2025-03-31
>
> We thank the reviewer for their time and valuable feedback. We are happy to respond to the questions that have been raised.
>
> **Extended experiments with CelebA.** We agree with the reviewer that extending experiments to cover other large datasets is interesting. In the graph (https://imgur.com/a/w89mL6I) we demonstrate generation of CelebA images from noise, following the experimental protocol of IGN, adjusted to use our Modified Backpropagation method. We train the same U-Net DCGAN architecture in the way outlined in our paper. Hyperparameters $\lambda$ were chosen as $\lambda_r = 20$, $\lambda_i = 0.006$, $\lambda_t = 0.02$ by trial and error. As with MNIST (and IGN), we sample noise with similar frequency-statistics as images from the dataset.
>
> Although the results are qualitatively inferior to state-of-the-art generative models, we believe that this is partly due to suboptimal hyperparameter selection, which would likely be improved with further fine-tuning. Nevertheless, the provided graph clearly demonstrates similar behaviour to that observed in our paper for the MNIST dataset (Figures 8 and 9), and in the IGN paper for MNIST and CelebA (Figure 4.) In particular, the images $f(\mathbf{z})$ and $f(f(\mathbf{z}))$ are highly similar as one expects from an idempotent mapping. Furthermore, we also observe the desired self-correcting property, with some small defects in background, hairstyle, and facial features visible in the image $f(\mathbf{z})$ then being corrected in the image $f(f(\mathbf{z}))$.
>
> A key result of the IGN paper is the ability to correct degraded images and map them into the distribution. In the graph (https://imgur.com/a/j4d20Ni) we demonstrate how a model trained using our method can correct images from the CelebA dataset after applying added noise, a greyscale filter, and a sketch filter. As in IGN, or model does not perfectly recover the original image, but characterizing features are clearly recovered in many cases.
>
> The results on CelebA were not initially included in the paper as we wish to focus primarily on the task of finding general idempotent mappings, as opposed to focusing on generative applications of the method. However, as experiments on CelebA was requested by several reviewers, we propose to include a short section on these results in the final paper, if our work is accepted.
>
> **Figures 3-7 in larger-scale settings.** The larger-scale generative setting we consider in this paper has a loss function which is composed of several components, some of which are adversarial in nature. It is well known that training generative models requires careful balancing of the hyperparameters in order to achieve good qualitative results, and it is rarely beneficial to let the idempotent loss factor too greatly in the combined loss. On that basis, even if our approach would be able to find models with lower idempotent loss, we believe that reproducing Figures 6-7 in this setting would be misleading, as they cannot be used meaningfully to judge the quality of the trained network.
>
> Figures 3-5 could indeed be reproduced meaningfully for our MNIST dataset, and we agree with the reviewer that it would be interesting to see whether the behaviour we observe on toy networks also applies to the larger-scale setting. We plan to do this thoroughly in future work, where we can give a fuller analysis of our method for more complex architectures.

---

> > ### Comment · Reviewer_hBam · 2025-04-01
> >
> > I thank authors for the detailed response. I'll keep my score.

---

### Official Review · Reviewer_kYga · 2025-03-14

**Overall Recommendation:** 2

**Summary:**

The paper presents a new approach to enforcing idempotency in neural networks through a modification of the backpropagation algorithm, termed Modified Backpropagation. The key idea is the derivation of an idempotent corrector function $g(K) = 3K^3 - 2K^2$, which iteratively projects a real-valued matrix onto the manifold of idempotent matrices. The authors extend this idea to general neural network training by modifying the canonical gradient-based optimization approach. Experimental results demonstrate that this method reduces idempotent error and outperforms ordinary backpropagation in multiple MLP and CNN architectures.

**Claims And Evidence:**

The claims made in the submission supported by clear and convincing evidence.

**Essential References Not Discussed:**

I believe that most relevant works have been cited in the paper.

**Experimental Designs Or Analyses:**

Yes.

**Methods And Evaluation Criteria:**

The proposed method makes sense for the problem and its application.

**Other Comments Or Suggestions:**

No.

**Other Strengths And Weaknesses:**

**Strengths**

- The paper is well-written and it is easy to follow.

- The introduction of an idempotent correction function and its integration into neural network training is a promising research topic as it has not been explored deeply in the literature. The relation with perturbation theory is interesting.

- The paper includes experiments demonstrating improved idempotent properties in trained networks compared to ordinary backpropagation.

- The method has been applied to generative models.

**Weaknesses**

- The theory part of the paper is weak. The relation of the proposed method to the perturbation theory as well as the stability analysis is quite easy and straightforward.

- The paper primarily focuses on MLPs and CNNs, but it does not provide results on more complex architectures like transformers or diffusion models.

- The computational cost of enforcing idempotency through Modified Backpropagation is not discussed in detail. Does the method introduce significant overhead compared to traditional approaches?

- It is unclear how much each component (e.g., the choice of recurrence relation, hyperparameter γ) contributes to the observed improvements.

**Questions For Authors:**

- How does the proposed method generalize to architectures beyond MLPs and CNNs, such as transformers or graph neural networks?

- In Figure 6-B1, why the error of the modified backpropagation it not stable. Is there any reason why?

**Relation To Broader Scientific Literature:**

This paper provides a new way to enforcing idempotency in neural networks through a modification of the backpropagation algorithm.

**Theoretical Claims:**

I have checked all details of the theory part. There is no actual proof.

---

> ### Author Rebuttal · Authors · 2025-03-31
>
> We thank the reviewer for their time and valuable feedback. We appreciate the concerns raised, and we aim to address these below.
>
> **Theoretical development.** We employ a novel theoretical framework which allows gradient-free training of an idempotent property. For us it is interesting that there exists a unique order-3 polynomial iterator $\mathbf{K}' = 3 \mathbf{K}^2 - 2 \mathbf{K}^3$ for the idempotence property. While the stability analysis may be technically straightforward, we have not seen perturbative techniques used for similar purposes in the machine learning community, and we believe our methods could be extended to other algebraic structures beyond idempotence. We therefore suggest that our work could be of theoretical interest to others in the community.
>
>
> **Computational cost.** A theoretical analysis shows that ordinary backpropagation and modified backpropagation have the same computational complexity. These results are reflected in practical experiments (https://imgur.com/a/S8FSExJ), which show that both algorithms run in approximately the same amount of time. These results show, for networks B1-B4 in Table 1, average wall-clock running time consumed per training epoch across 250 repetitions of each algorithm, with 98\% confidence intervals. Across all configurations, both algorithms take approximately the same amount of time, with a slight advantage to Modified Backpropagation with relative improvement ranging from $1-38\%$.
>
> The theoretical argument goes as follows. We here consider the growth in the number of matrix multiplications required as the number of layers increases. Defining $y=f(\mathbf{x})$, the loss function from Eq 2 can be written as $L(\mathbf{y}) = \frac{1}{m} \sum (f(\mathbf{y}) - \mathbf{y})^2$. By the chain rule $\frac{\partial{L}}{\partial{\mathbf{W}}} = \frac{\partial{L}}{\partial{\mathbf{y}}} \frac{\partial{\mathbf{y}}}{\partial{\mathbf{W}}}$. Computing $\frac{\partial{\mathbf{y}}}{\partial{\mathbf{W}}}$ using backpropagation will generally use $O(k)$ matrix multiplications for a $k$-layer MLP. This quantity is computed in the same way for both Modified Backpropagation and Ordinary Backpropagation.
>
> For Ordinary Backpropagation, the quantity $\frac{\partial{L}}{\partial{\mathbf{y}}}$ can also be unfolded via the chain rule and its evaluation requires computing $\mathbf{y} = f(\mathbf{x})$, $f(\mathbf{y})$, as well as $\frac{\partial{f(\mathbf{y})}}{\partial{\mathbf{y}}}$, which each can be computed in $O(k)$. Thus, assuming memoization is used (as is the case by design in most autodiff frameworks, like PyTorch), Ordinary Backpropagation can compute $\frac{\partial{L}}{\partial{\mathbf{W}}}$ in $O(k)$ time. (In Section 1 around Eq. 2 in the paper we argue that the computational cost of evaluating this gradient in Ordinary Backpropagation grows exponentially in the number of layers. Memoisation reduces this to $O(k)$ time, and we propose to clarify this point in the paper.) For Modified Backpropagation we compute $\frac{\partial{L}}{\partial{\mathbf{y}}} = 3f(\mathbf{y}) - 2f(f(\mathbf{y})) - \mathbf{y}$. Again, assuming memoization, each of these terms take $O(k)$ time to evaluate, so Modified Backpropagation also runs in $O(k)$ time.
>
> In the final paper, if accepted, we will include content on the theoretical discussion on relative running time, as well as the graph showing our practical experiments which demonstrate this point.
>
>
> **Error spikes in Figure 6-B1.** The spikes are caused by some runs of Modified Backpropagation converging (within machine precision) to the zero-matrix during training. Due to floating-point imprecision this is rounded down and causes the effect. We chose not to exclude these runs from analysis as we deem it important to expose this behaviour. Note that this only occurs on the small architecture B1. The last sentence in the caption for Figure 6 addresses exactly this concern. In the final paper, if accepted, we will add a further sentence in the main body to make this point more clearly.
>
>
> **Application to other network architectures.** We appreciate the relevance of exploring more complex network architectures. This work proves the principle that a fundamentally new approach to learning idempotent networks is possible. In this early work it has been important to explore behaviour of Modified Backpropagation in controlled environments with few external factors in order to give convincing evidence of the method's efficacy. To this end, we have evaluated our method on a variety of MLP networks and synthetic datasets which illuminates behaviour such as sensitivity to network size, dataset distributions, and running time. Nevertheless, wider application of the method in practice will depend on generalization to the architectures mentioned by the reviewer, and we are eager to explore this in future work.

---

### Decision · Program_Chairs · 2025-05-01

**Decision:**

Accept (poster)

**Comment:**

The paper studies a modified backpropagation method based on perturbation theory to enforce idempotency (f(f(x)) = f(x)) in generative neural networks. The majority of reviewers agree that the approach is novel and elegant, and experiments support that the idempotency error improves across various neural network architectures. However, some reviewers point out that the experimental scope is somewhat limited, e.g., lacking evaluation on more complex models (Transformers, diffusion models) or larger, more diverse datasets (e.g., CelebA). Focusing on the experimental part would definitely make the paper stronger.

In addition to reviewers' comments there are several inaccuracies which need to be fixed in the final version:

- After equation (4) it is stated that it leads to a linear program (which is actually a linear program with quadratic constraints (with terms of the form \alpha_i \alpha_j for any i,j -- including i=j). Does this affect derivations?

- Around (6) d_\lambda is defined as the multiplicity of an eigenvalue, but it is rather the size of the corresponding Jordan block. This leads to stating that "geometric multiplicity of every eigenvalue must also be exactly 1", which is clearly incorrect (excluding, e.g., the identity matrix, which is clearly a solution).